# The mechanism for the enhanced piezoelectricity in multi-elements doped (K,Na)NbO$_3$ ceramics

Xiaoyi Gao[1,2,3], Zhenxiang Cheng [2], Zibin Chen [4], Yao Liu[5], Xiangyu Meng[1], Xu Zhang[1], Jianli Wang [2], Qinghu Guo[3], Bei Li[1], Huajun Sun[3], Qinfen Gu [6], Hua Hao[3], Qiang Shen [1✉], Jinsong Wu [1✉], Xiaozhou Liao [4], Simon P. Ringer [4], Hanxing Liu[1,7], Lianmeng Zhang[1], Wen Chen[1], Fei Li [5✉] & Shujun Zhang [2✉]

(K,Na)NbO$_3$ based ceramics are considered to be one of the most promising lead-free ferroelectrics replacing Pb(Zr,Ti)O$_3$. Despite extensive studies over the last two decades, the mechanism for the enhanced piezoelectricity in multi-elements doped (K,Na)NbO$_3$ ceramics has not been fully understood. Here, we combine temperature-dependent synchrotron x-ray diffraction and property measurements, atomic-scale scanning transmission electron microscopy, and first-principle and phase-field calculations to establish the dopant–structure–property relationship for multi-elements doped (K,Na)NbO$_3$ ceramics. Our results indicate that the dopants induced tetragonal phase and the accompanying high-density nanoscale heterostructures with low-angle polar vectors are responsible for the high dielectric and piezoelectric properties. This work explains the mechanism of the high piezoelectricity recently achieved in (K,Na)NbO$_3$ ceramics and provides guidance for the design of high-performance ferroelectric ceramics, which is expected to benefit numerous functional materials.

[1] State Key Laboratory of Advanced Technology for Materials Synthesis and Processing, School of Materials Science and Engineering, Wuhan University of Technology, Wuhan, China. [2] Institute for Superconducting and Electronic Materials, Australian Institute for Innovative Materials, University of Wollongong, Wollongong, NSW, Australia. [3] State Key Laboratory of Silicate Materials for Architectures, Center for Smart Materials and Device Integration, Wuhan University of Technology, Wuhan, China. [4] School of Aerospace, Mechanical and Mechatronic Engineering, The University of Sydney, Sydney, NSW, Australia. [5] Electronic Materials Research Laboratory, Key Laboratory of the Ministry of Education and International Center for Dielectric Research, Xi'an Jiaotong University, Xi'an, China. [6] Australian Synchrotron (ANSTO), Clayton, Australia. [7] International School of Materials Science and Engineering, Wuhan University of Technology, Wuhan, China. ✉email: sqqf@whut.edu.cn; wujs@whut.edu.cn; ful5@xjtu.edu.cn; shujun@uow.edu.au

Piezoelectric materials, which convert mechanical energy to electrical energy or vice versa, are at the heart of numerous electromechanical applications, such as piezoelectric actuators, ultrasonic medical imaging, structural health monitoring and mechanical energy harvesting, to name a few[1–3]. To improve the performance of electromechanical devices, the selection of a piezoelectric material with optimized properties is critical. Among the known piezoelectric materials, perovskite ferroelectric materials, such as $Pb(Zr,Ti)O_3$ (PZT) and relaxor-$PbTiO_3$ based solid solutions, have exhibited the highest piezoelectric coefficient and electromechanical coupling factors due to the existence of morphotropic phase boundary (MPB) and the ability to tailor properties between "soft" and "hard" piezoelectric responses[1–4]. As a result, they have been the mainstay piezoelectric materials and have occupied the largest share of the piezoelectric ceramics market.

The manufacturing and applications of lead-based materials, however, have raised concerns from the increased awareness and regulations regarding the health and environmental issues associated with the use of lead throughout their life cycles[5–7]. Thus, the studies on lead-free piezoelectric materials have resurged in the past 20 years. In recent years, the potassium sodium niobate (KNN) based system has become a leading candidate as a lead-free alternative with gradually increasing piezoelectric properties[7–9]. Generally, the enhanced piezoelectric activity in KNN ceramics has been attributed to the polymorphic phase transition (PPT) shifting downward to room temperature, achieved by doping various elements, such as $Li^+$, $Sb^{5+}$, $Ta^{5+}$, etc[6–12]. To further improve the piezoelectric performance, recent efforts have focused on the construction of MPBs by adding appropriate dopants, thus achieving much-improved piezoelectricity in multi-elements doped KNNs, with piezoelectric coefficients ($d_{33}$) being above 500 pC/N and comparable to "soft" PZTs[11–13]. According to the phenomenological theory, in regions with multiple coexisting phases (such as PPT or MPB regions), both intrinsic lattice distortion and extrinsic domain wall motion can be enhanced because the energy barriers between the various ferroelectric phases and the curvature of the free energy profile with respect to polarization at stable states are minimized[13–17].

Notably, the piezoelectric coefficients $d_{33}$ of single element doped KNN ceramics are generally <200 pC/N[18,19], which is much lower than that of their multi-elements doped counterparts. Considering that the multi-elements doped KNN-based ceramics have exhibited good thermal stability with higher piezoelectric properties at a temperature above PPT[12,20–22], this enhanced piezoelectricity cannot be explained by the concept of the PPT. Until now, the impact of dopants on the local structure heterogeneity, average structure and piezoelectricity in KNN-based ceramics has not been fully understood, which has significantly hindered the development of lead-free piezoelectric ceramics. To explore the underlying mechanisms of high piezoelectricity in multi-elements doped KNN based ceramics, we conducted comparative investigations on Bi,Sb,Zr multi-elements doped KNN (KNN-Bi,Sb,Zr) and single element doped ones, including Bi- (KNN-Bi), Sb- (KNN-Sb), and Zr- (KNN-Zr) doped KNNs. We demonstrate the connections between composition fluctuation, microstructure heterogeneity and macroscopic properties, and provide inspiration for the design of lead-free ferroelectrics with high piezoelectricity and thermal stability.

## Results

The major properties and field-emission scanning electron microscopic (FE-SEM) images of the KNN-Bi,Sb,Zr sample are given in Supplementary Table 1 and Supplementary Fig. 1 respectively, which are compared with those of the single element doped KNNs. It is found that the KNN-Bi,Sb,Zr exhibits the highest dielectric and piezoelectric properties with values of 2900 and 520 pC/N at room temperature, respectively. These values are much higher than those of the single element doped counterparts, which are below 1000 and 200 pC/N, respectively. The two-step sintering procedure (the details are given in the Materials Synthesis) was employed for the multi-elements doped KNN, where large grain size on the order of 25 μm was obtained at optimized sintering temperature, much larger than those of single element doped counterparts prepared by conventional one-step sintering process, though all of the samples have a similar density of 94-96%. To explore the intrinsic and extrinsic contributions of the piezoelectricity, we performed Rayleigh analysis[23,24]. Supplementary Fig. 2 shows $d_{33}$ value versus the amplitude of a.c. electric field for the studied KNN ceramics, where the applied electric field is below the half of the coercive field (~being on the order of 8 kV/cm for KNN-Bi,Sb,Zr, as given in Supplementary Fig. 2a). We found that the $d_{33}$ showed a linear behaviour with respect to the driving field, where the Rayleigh parameters $d_{init}$ (intrinsic contribution) and $\alpha$ (contribution from irreversible domain wall motion) were found to be 460 and 72 cm/kV for KNN- Bi,Sb,Zr, respectively, both of which were greatly enhanced compared with its single element doped counterparts, being ~80–120 and 10 cm/kV, respectively. Thus, the high piezoelectricity in multi-elements doped KNN can be attributed to both the enhanced intrinsic piezoelectricity and the contribution from domain wall motion.

To explore the contribution of multi-elements doping on the average structure, we performed synchrotron X-ray diffraction (XRD) to study the structure evolution of the doped KNN ceramics over a temperature range of 123–423 K, as shown in Supplementary Figs. 3–6. The enlarged main diffractions, including (100)/(200), (110), and (111), as a function of temperature are given in Fig. 1a, b, whereas the phase fractions are represented in Fig. 1c, d. Both KNN-Sb and KNN-Bi,Sb,Zr possess the orthorhombic (O) and tetragonal (T) phases over the studied temperature range. At room temperature, the dominate phase of KNN-Bi,Sb,Zr ceramics is a T phase with a volume fraction of >95%, whereas the O phase dominates the KNN-Sb ceramics with T phase being <30%. The increase in volume fraction of the T phase in doped KNN ceramics indicates that the dopant can induce an average structure transition from the O phase to the T phase.

To further confirm this phase evolution trend, we performed first-principle calculations based on density functional theory to optimize the structure of Bi, Sb and Zr doped KNNs and shown in Fig. 1e. It should be noted that the additions of $Bi^{3+}$ and $Zr^{4+}$ to A- and B- sites of KNN ceramics will inevitably introduce vacancy defects such as A-site vacancies and oxygen vacancies, respectively. The contributions of the K vacancy and oxygen vacancy to the structure were also studied based on first-principle calculations and given in Supplementary Fig. 7. Interestingly, the difference in the six B–O bond lengths of the octahedron, after adding dopants (Fig. 1e) and oxygen vacancy, is obviously reduced compared with the undoped KNN, approaching the same value. It should be noted that after adding K vacancy, however, the lengths of the six B–O bonds of the octahedron are still different, though the K vacancy can indeed slightly change the bond lengths of the octahedron comparing to those of the pure KNN (Supplementary Fig. 7), demonstrating that the loss of alkali metal has relatively small impact on the octahedrons of KNN. This result confirms the significant reduction of the octahedral distortion, thus the phase evolution trend towards highly symmetrical pseudo-cubic is expected regardless of the dopant types, either on the A sites or B sites.

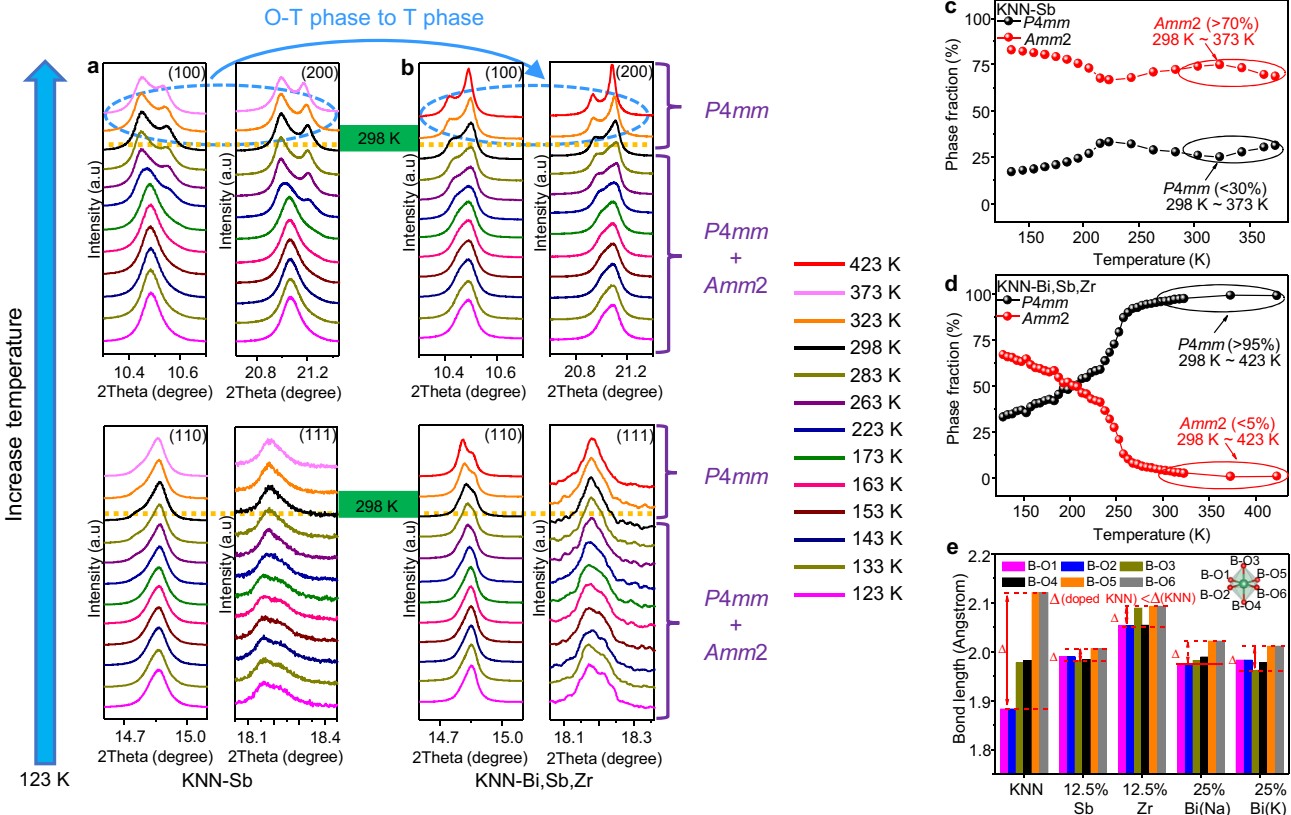

**Fig. 1 Structural evolutions of the KNN-Bi,Sb,Zr and KNN-Sb ceramics. a**, **b** Detailed profiles of main diffractions including (100)/(200), (110), and (111) as a function of temperature for KNN-Sb and KNN-Bi,Sb,Zr ceramics, respectively. **c**, **d** The phase fraction as a function of temperature for KNN-Sb and KNN-Bi,Sb,Zr ceramics, respectively. **e** The lengths of the six B–O bonds of perovskite octahedron for undoped and single element doped KNNs obtained from first-principles calculations.

To investigate the average structural-property relationships, we measured the dielectric and piezoelectric properties as a function of temperature for both multi-elements and single-element doped KNNs. Figure 2, Supplementary Figs. 8 and 9 show the temperature-dependent dielectric and piezoelectric properties for the doped KNN ceramics. Figure 2a gives dielectric properties of KNN-Bi,Sb,Zr ceramics over a temperature range of 130–700 K, with dielectric anomalies at 550 K and 290 K, which correspond to Curie temperature and O–T phase transition temperature (Fig. 1d), respectively. It should be noted that the tetragonal phase still exists above the Curie temperature over a broad temperature range even the volume fraction is well below 10% (Supplementary Fig. 10), which can also be confirmed by the broad dielectric peak at 550 K as shown in Fig. 2a. Both temperatures are greatly reduced when compared with the single-element doped counterparts (Supplementary Fig. 8), which also confirm the results of synchrotron XRD and first-principle calculations, that is, the reduction of the octahedral distortion and the phase evolution trend towards highly symmetrical pseudo-cubes. Of particular importance is that the permittivity of KNN-Bi,Sb,Zr ceramic increases from 380 to 2900 over the cryogenic temperature of 130–300 K, whereas the permittivity increment is below 700 for its single doped counterparts over the same temperature range, as shown in Fig. 2b. As expected, the temperature-dependent piezoelectric properties of KNN-Bi,Sb,Zr ceramic also exhibit a large enhancement over the cryogenic temperature range compared with its single doped counterparts (Fig. 2c). The piezoelectric coefficient $d_{33}$ of KNN-Bi,Sb,Zr is still high at elevated temperature, which is >360 pC/N at 423 K, where the average structure is tetragonal, as shown in Fig. 2d. Therefore the high

piezoelectric properties of KNN-Bi,Sb,Zr ceramic are not only associated with the coexistence region of different ferroelectric phases at MPB or PPT[6–16,25], but also governed by the average tetragonal phase structure. To understand the mechanism responsible for the enhanced properties, we measured the relative dielectric permittivity and loss factor as a function of frequency for KNN-Bi,Sb,Zr ceramic, as shown in Supplementary Fig. 11. The relative dielectric permittivity and loss factors over the temperature range from 153 to 293 K were found to be frequency-dependent, revealing a strong relaxation behaviour over the frequency range of $10^{0}$–$10^{6}$ Hz, demonstrating that a considerable portion of the dielectric permittivity is from the switching of specific dipoles and/or interface. This switching is believed to be associated with the local structural heterogeneity induced by the dopants[26–29]. Interestingly, the temperature range of the broad smeared relaxation peak of the loss factor in KNN-Bi,Sb,Zr can be represented by the temperatures of the loss relaxation peaks of single element doped ones, as shown in Supplementary Fig. 9, indicating that the local structural heterogeneity in multi-elements doped KNN is contributed by different dopants, while the behaviour of each individual dopant is slightly different.

We conducted atomic-resolution scanning transmission electronic microscopy (STEM) analysis on the KNN-Sb and KNN-Bi,Sb,Zr ceramics to investigate the role of dopants in local and average structures[30]. The K/Na (A sites) cations and Sb/Nb (B sites) cations site occupations for KNN-Sb ceramic are given in Fig. 3a. Figure 3b, c show the STEM-high-angle annular dark-field (HAADF) images with A and B site intensity maps, respectively. The intensity of atomic columns in a STEM-HAADF image is proportional to the atomic number. The different

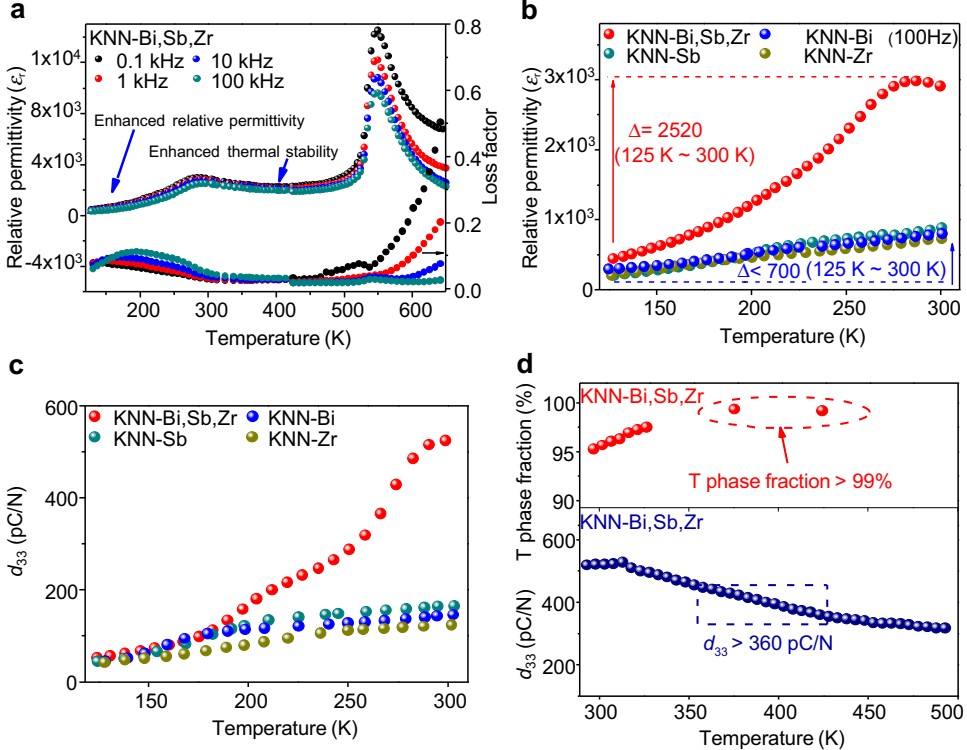

**Fig. 2 The dielectric and piezoelectric property evolutions for multi-elements and single-element doped KNN ceramics. a** The dielectric behaviour of KNN-Bi,Sb,Zr (multi-element doped KNN) ceramic over a temperature range of 130–700 K. **b** The dielectric and **c** piezoelectric properties of the multi-elements and single-element doped KNN ceramics over a temperature range of 120–300 K. **d** The temperature dependence of the tetragonal phase fraction and piezoelectric coefficients $d_{33}$ over a temperature range of 298–490 K for KNN-Bi,Sb,Zr ceramic.

intensities observed in Fig. 3b, c suggest the existence of strong local compositional fluctuations. Considering that the Na cation (atomic number 11) and K cation (atomic number 19) contributed to the $A$ site intensity, we attribute the brighter intensity areas to K cation enrichment. In contrast, for $B$ site intensity map, the intensity is related to the Nb cation (atomic number 41) and Sb cation (atomic number 51), and thus, we attribute the brighter intensity areas to the Sb cation enrichment. Figure 3d shows the polar vector map for KNN-Sb ceramics by measuring the displacement between K/Na ($A$ sites) cations and their surrounding Nb/Sb ($B$ sites) cations in the HAADF image. The KNN-Sb ceramics possess a combination of O and T phases based on synchrotron results, resulting in polarization directions pointing to [110] and [001] directions, respectively. In Fig. 3d, some areas possess polar vectors close to the [110] direction, whereas others approach the [001] direction, which confirms the coexistence of O and T phases. To explore the impact of elements on the local structure, we studied four distinct areas, including the Na/Nb rich (area I), K/Nb rich (area II), Na/Sb rich (area III), and K/Sb rich (area IV), and marked with solid ellipse accordingly in Fig. 3b, c, and d. The enlarged areas are represented in Fig. 3e, in which the angular differences between the directions of the polar vector and the [001] direction are marked with different colours. Interestingly, compared with the Na/Nb rich area, the polar vector arrangement in the K/Nb rich area is more ordered, and the average polar vector direction is close to the [110] direction, which reveals that the enrichment of the K cation favours the O phase. Compared with Sb deficient areas (areas I and II), the polar vector maps of Sb rich areas (areas III and IV) show more polar vectors with directions along the [001] direction (green boxes), whereas the others (marked with white circles among the chartreuse boxes in Fig. 4e) approach the [001] direction with significantly reduced deflection angles (the deflection angle is the

angle between the polar vector direction and the [001] direction). Thus, the Sb enrichment increases the volume fraction and size of the T phase with low-angle polar vectors (here we regard the deflection angle below 20° as a low-angle), leading to the average structure changing from the O phase to the T phase and is accompanied by large amount local structure heterogeneity.

Figure 3f shows a STEM-HAADF image with a polar vector map for KNN-Bi,Sb,Zr ceramics, exhibiting more areas (area V) with polar vectors along the [001] direction compared with that of the KNN-Sb ceramic, whereas many areas (area VI) possess polar vectors with low-angle close to the [001] direction, indicating that the overall structure approaches the T phase. We also confirmed this by synchrotron XRD as given in Fig. 1d, where the T phase is dominant at room temperature. Of particular significance is that the undulation of the polar vector direction in the KNN-Bi,Sb,Zr sample is much smaller compared with that of the KNN-Sb sample (Fig. 3d). A statistical representation of the polar vector angles and their proportions in KNN-Sb and KNN-Bi,Sb, Zr samples are given in Fig. 3g, indicating that the average T phase structure has more regions with low-angle polar vectors (area VI). The KNN-Bi,Sb,Zr ceramic contains more than 80% low-angle polar vectors whose deflection angle from [001] is <20°, whereas it is only 50% in KNN-Sb ceramic. Therefore, although the multi-elements doping induces the overall average structure from the O phase to the T phase, compared with the single-element doped counterparts, a larger amount of low-angle polar vectors are found in KNN-Bi,Sb,Zr ceramics. Thus, the deflection angle of the polar vectors decreases with an increase in the volume fraction of the tetragonal phase in the entire system, that is, as the enrichment of dopants increases or doping level increases, the polar vectors approach but are not along the [001] direction. This observation reveals that the dopants induce average T phase structure in KNN system, accompanied by large

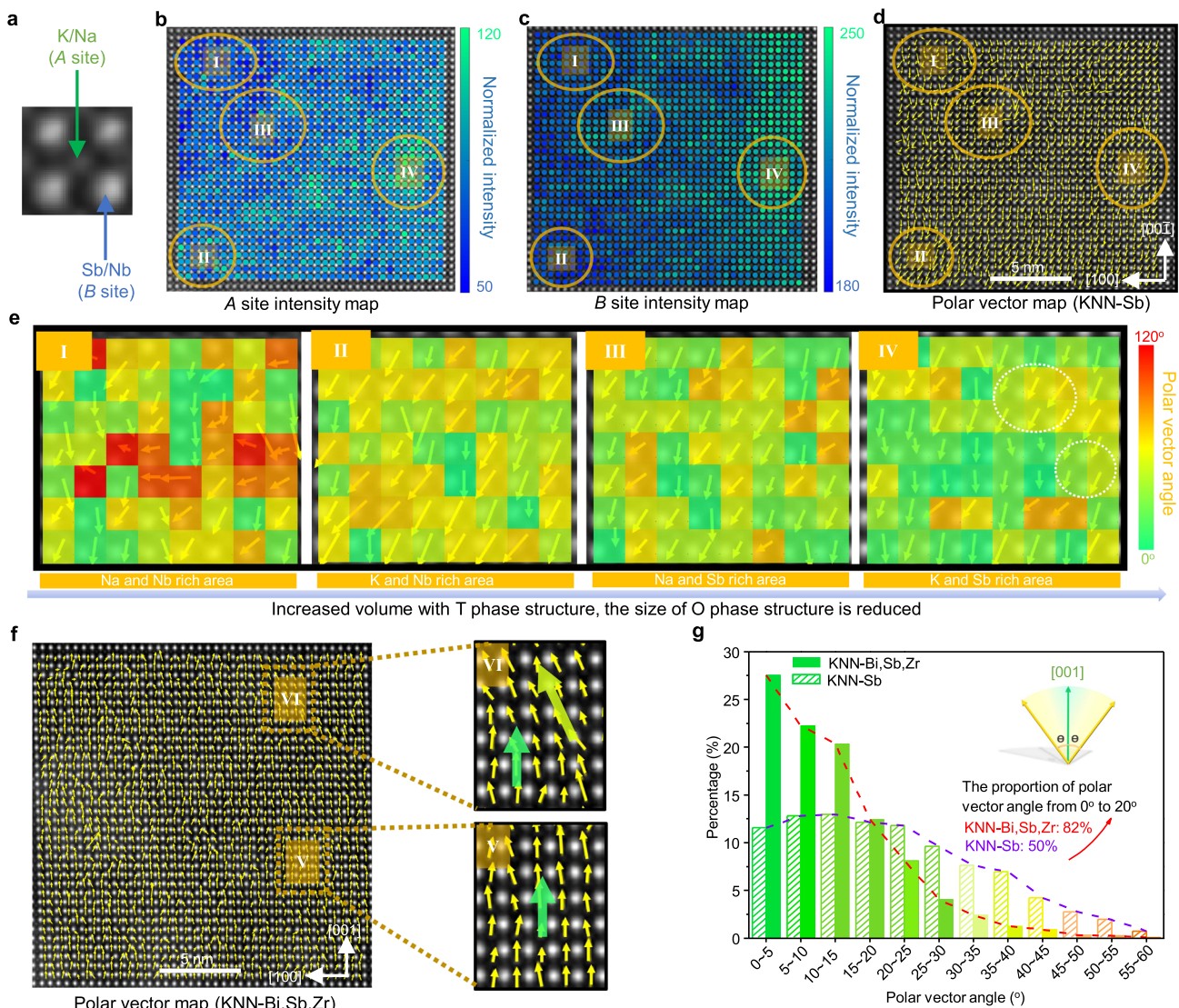

**Fig. 3 The relationship of dopant, average structure, and local structure in KNN-based materials. a** A unit cell showing the atomic site occupation for KNN-Sb ceramic. **b** Normalized intensity of the *A* site for the KNN-Sb ceramic. Areas with high *A* site intensity are K rich. **c** Normalized intensity of the *B* site for the KNN-Sb ceramic. Areas with high *B* site intensity are Sb rich. **d** Polar vector map for KNN-Sb ceramic, the polar vectors (arrows) are given for each unit cell column. A unit cell with the polar vector direction along the [001] direction is of the T structure, and along the diagonal direction is of the O structure. **e** The enlarged areas of Na/Nb, K/Nb, Na/Sb, and K/Sb rich in (**d**). The different colour of unit cells shows the angle of polar vector direction off the [001] direction. **f** Polar vector map for KNN-Bi,Sb,Zr ceramic. The enlarged areas show the local average polar vector directions are slightly off the [001] direction (area VI) and along the [001] direction (area V), respectively. **g** The statistic percentage of polar vectors as a function of the polar vector angle (i.e., the deflection angle of polar vector off the [001] direction), based on (**d**, **f**).

amount of low-angle local structure heterogeneity, as shown in Fig. 3e–g.

## Discussion

We performed phase filed simulations to understand the contribution and demonstrate the importance of the low-angle local structure heterogeneities to piezoelectricity[31,32], as shown in Fig. 4 and Supplementary Fig. 12. We compared the microstructure evolution of the pure T phase and the T phase containing local orthorhombic regions (volume fraction is 10%) as a function of the applied electric field. As expected, the average polarization of the T phase containing local orthorhombic regions easily rotates under the external electric field, indicating that the local structure heterogeneity can facilitate the polarization rotation and, therefore, enhance the dielectric and piezoelectric properties. Of particular importance is that

the low-angle polar vectors (light yellow and light blue areas in Fig. 4b at 0 kV/cm) and nearby structure change substantially with an increase in the external electric field. In contrast, the local structures with large-angle polar vectors (i.e., the deflection angle deviating from the tetragonal spontaneous polarization direction [001] is >20°) under the external electric field change much smaller (the orange frame regions in Fig. 4a) when compared to the local structures with low-angle polar vectors, revealing the polar vectors with large-angles response weakly to the external electric field. These different responses to the external electric fields between the local structures with large- and low-angle polar vectors result from the competition between the Landau energy barrier and the interfacial energies. The interfacial energies, including electrostatic, elastic, and gradient energies, are generated due to the discontinuity of the polarization and strain[28].

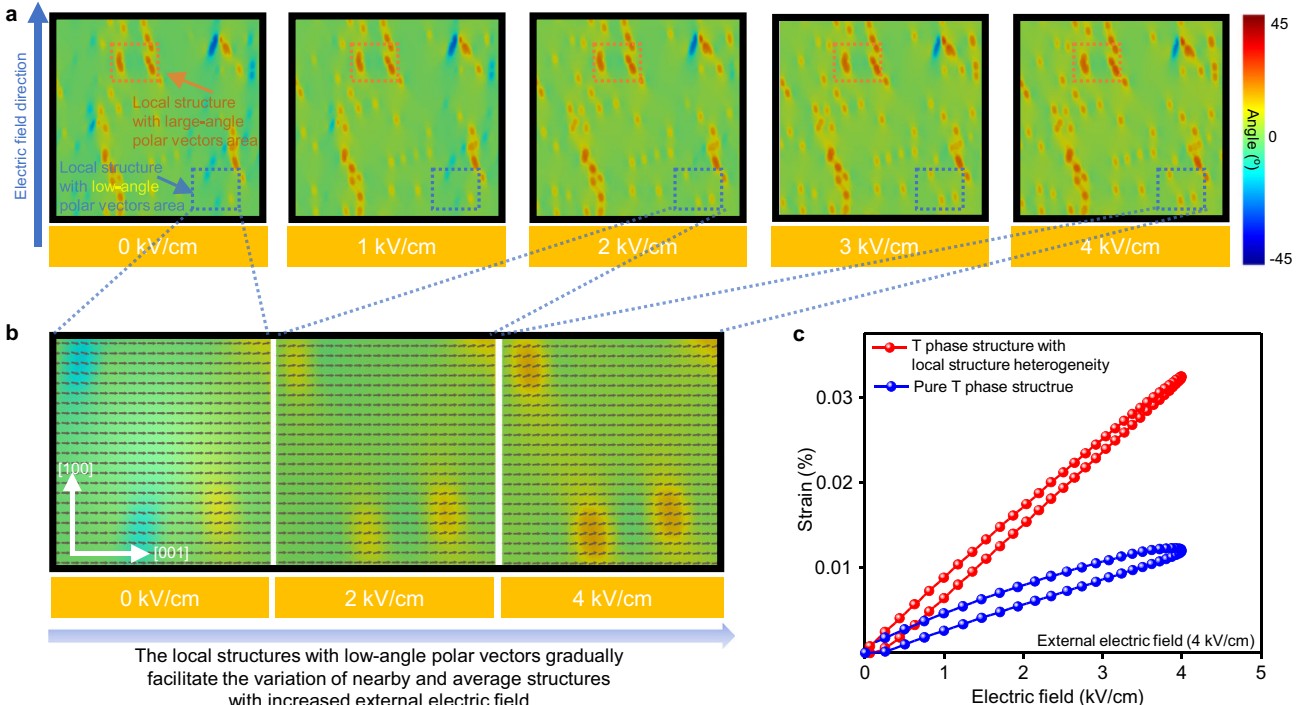

**Fig. 4 Phase-field simulation results of mesoscale microstructure and piezoelectric strain. a** Microstructural evolution as a function of applied electric field along the [100] direction for the average T phase with local structure heterogeneities. The colour bar shows the angle of the polar vector direction off the [001] direction. Here, the green colour regions (angle is 0°) show the polar vector direction is along the [001] spontaneous polarization direction of the T phase. The regions with other colours show the polar vector directions are gradually deviated from the [001] direction. The local structures with large-angle polar vectors are marked with an orange frame. **b** The enlarged areas of the local structures with low-angle polar vectors in (**a**). The black arrows show the polar vectors. **c** The simulated piezoelectric strain under the electric field at 4 kV/cm for tetragonal phase with and without local structure heterogeneities.

For low-angle polar vectors, the Landau energy and interfacial energies are similar, leading to a flat local free energy profile, making them highly instable and is easier for polar vector rotation. For large-angle polar vectors, on the contrary, the impact of interfacial energy is much smaller, and therefore these regions are close to their primitive orthorhombic phase which is controlled by its Landau energy, where the polar vectors are difficult to rotate when compared to the low-angle counterpart. As shown in Fig. 3, the deflection angle of local structure heterogeneity is strongly associated with the volume fraction of the tetragonal phase. Thus, this explains the high piezoelectricity in KNN-Bi,Sb,Zr ceramics, which have an average T phase structure and a large amount of nanoscale local structures with low-angle polar vectors, as confirmed by STEM and synchrotron XRD results. Figure 4c and Supplementary Fig. 13 present the calculated and experimentally measured piezoelectric strain curves of the pure tetragonal phase and the tetragonal phase with local structure heterogeneity, respectively. As expected, the electric-field-induced strain is three times higher for the system with local structure heterogeneity.

As discussed earlier, the contribution of multi-elements doping on structures and piezoelectricity can be attributed to the following factors: (1) the presence of dopant-induced tetragonal phase and (2) nanoscale local structure heterogeneities with low-angle polar vectors. For perovskite ferroelectric ceramics, the highest piezoelectricity is generally present on the tetragonal side of MPB compositions, such as PMN-PT and PZT ceramics[31,33,34]. We believe that this is due to the piezoelectric anisotropy characteristics of the perovskite ferroelectrics. That is, the $d_{33}$s could be at high levels for all orientations in the tetragonal perovskite, while in orthorhombic and rhombohedral perovskites, the $d_{33}$s are low in multiple directions, which deteriorate the average piezoelectric property in polycrystalline

ceramics with O and R phases[35]. In addition, the high mobility of the 90° domain walls and hierarchical domain structure (Supplementary Fig. 14) in the large grain of KNN-Bi,Sb,Zr ceramics may also be an important contributor to the enhanced extrinsic piezoelectricity[22,33,34]. The existence of low-angle polar regions that induced by multi-elements doping can further promote the intrinsic contribution, i.e., the polarization rotation process, because these nanoscale local structures are sensitive to external electric fields as a result of the energy competition between the local Landau energy and interfacial energy. Thus, the piezoelectric property is enhanced with increased volume of the average tetragonal phase structure accompanied by a large amount local structure heterogeneities. For single-element doped KNNs, however, because of the solubility limitation, the dopant concentration is not sufficient to induce an average tetragonal phase. Meanwhile, the percentage of low-angle polar vectors is much lower compared with the multi-elements doped KNN ceramic. Therefore, to achieve high piezoelectricity in KNN ceramics, we have to use multiple dopants to break the solubility limit of the single dopant and to transform orthorhombic KNN into average tetragonal KNN. This also introduces high-density nano-regions with low-angle polar vectors because of the different ionic radii and valences of the dopants, which couple with the tetragonal phase and synergistically contribute to the intrinsic contribution of the high piezoelectricity in KNN-based lead-free ceramics. It should be noted here that $Bi^{3+}$ doping cation on A-site of KNN and $Zr^{4+}$ doping cation on B-site KNN ceramics will induce point defects, i.e., K vacancy and oxygen vacancy respectively, based on the following defect reactions:

$$Bi_2O_3 \xrightarrow{K_{0.5}Na_{0.5}NbO_3} 2Bi_K^{\cdot\cdot} + 4V_K' + 3O_O \quad (1)$$

$$2ZrO_2 \xrightarrow{K_{0.5}Na_{0.5}NbO_3} 2Zr'_{Nb} + V_O^{\cdot\cdot} + 4O_O \qquad (2)$$

Meanwhile, the $Sb^{5+}$ doping cation on B-site of KNN ceramic will keep charge balance without introducing the point defects. On the other hand, in our studied multi-elements doped KNN, we attempted to keep the charge balanced by tuning the composition with the appropriate doped levels, thus the formation of point defects is not required.

Interestingly, the rare-earth-doped PMN-PT ceramics with enhanced piezoelectric coefficient also exhibit increased volume fraction of tetragonal phase with reduced Curie temperature and broad relaxation peak[31,36]. Tetragonal phase with low-angle domain wall was also observed in the structure evolution of PMN-PT solid solution with increasing Ti concentration based on the molecular dynamic calculation and STEM observation, being considered responsible for its high piezoelectric properties[26,37,38].

In summary, the evolutions of the local structure heterogeneity and average tetragonal structure with increased dopant level in KNN based ceramics are revealed in this research, which can explain the underlying mechanism responsible for the high piezoelectricity recently discovered in KNN ceramics. We believe that the dopants induced average tetragonal phase structure with plentiful local structure heterogeneities is not unique in KNN lead free but is general in ferroelectric ceramics, which can be employed as a guideline to further increase the density of local structure heterogeneities with comparable Landau energy and interfacial energy, thus improve the piezoelectric properties in numerous ferroelectric ceramics.

## Methods

**Materials synthesis**. We weighed and mixed the high-purity raw materials $Na_2CO_3$ (99.6%, Alfa Aesar), $K_2CO_3$ (99.0%, Alfa Aesar), $Bi_2O_3$ (99.975%, Alfa Aesar), $Sb_2O_3$ (99.9%, Alfa Aesar), $ZrO_2$ (99.99%, Aladdin), and $Nb_2O_5$ (99.5%, Alfa Aesar) according to the nominal compositions of $K_{0.5}Na_{0.5}Sb_{0.04}Nb_{0.96}O_3$ (KNN-Sb), $K_{0.48}Bi_{0.02}Na_{0.5}NbO_3$ (KNN-Bi), $K_{0.5}Na_{0.5}Zr_{0.01}Nb_{0.99}O_{2.995}$ (KNN-Zr), and $K_{0.48}Bi_{0.02}Na_{0.5}Nb_{0.92}Sb_{0.04}Zr_{0.04}O_3$ (KNN-Bi,Sb,Zr), respectively. The mixed powders were calcined at 850 °C for 6 h and then milled in anhydrous ethanol ($CH_3CH_2OH$ >99.5%, 200 proof, Sigma Aldrich). Fine powders with PVA were compacted into disks by uniaxial pressing at 25 MPa for 2 min. Finally, the KNN-Bi, KNN-Sb, and KNN-Zr pellets were sintered at 1130 °C–1180 °C for 2–4 h. The KNN-Bi,Sb,Zr pellets were prepared by two-step sintering method, heating up to 1190 °C at a rate of 10 °C/min, then cooled down to 1090 °C at a rate of 10 °C/min and kept for 10 h. The grain size was measured by Nano Measurer software based on the images obtained by a field-emission scanning electron microscopy (FESEM, FEI Quanta FEG250). The densities of the ceramics were measured by the Archimedes method, where the relative density was calculated based on the theoretical density of KNN[39].

**Structure characterization by synchrotron radiation X-ray diffraction (synchrotron XRD)**. We performed the in-situ synchrotron XRD experiments on the KNN-Bi,Sb,Zr, and KNN-Sb ceramics powders at the Australian Synchrotron Beamline from 423 K down to 123 K. The samples were dried at 180 °C for 2 h before the test. To avoid the effect of fluorescence, we set the wavelength λ and the instrument resolution Δd/d to 0.727464 Å and 0.00375, respectively. The in-situ XRD experiments on the KNN-Bi,Sb,Zr, ceramics powders at a temperature from 500 K to 800 K was performed by XRD (SmartLab, Rigaku, Japan) operating at 40 kV and 200 mA and using Cu Kα1 radiation. We used FULLPROF software to perform the Rietveld refinements. The values of full width at half maximum stand for the entire broad peak, derived from peak analysis using the Origin software built-in package. The database CIF files were ICSD 173741, ICSD 253347, and ICSD 186364 for $P4mm$, $Amm2$, and $Pm-3m$, respectively.

**First-principles calculations**. We performed calculations based on density functional theory (DFT) using the Vienna ab-initio simulation package (VASP)[40]. For the exchange-correlation functional, we used the generalized gradient approximation (GGA) of Perdew-Burke-Emzerhof for solid (PBESol)[41]. We described electron-ion interactions using the projector augmented-wave (PAW) potential with a kinetic energy cut-off of 500 eV. The K 3s3p4s, Na 2p3s, Bi 5d6s6p, Nb 4s4p4d5s, Sb 5s5p, Zr 4s4p4d5s, and O 2s2p states were treated as valence electrons. We performed structure optimization of $2 \times 2 \times 2$ 40-atoms pseudocubic supercell of undoped, doped, and vacancy-defect KNN models. The lattice

parameters ($a = c > b$) of the pseudocubic primitive cell correspond to the [$\bar{1}01$], [010] and [101] directions of the $Amm2$ space group, which is the typical phase of KNN at room-temperature[42]. The supercell with the uniform distribution of A-site cations along the a, b, and c directions was used in the calculations, as shown in Supplementary Fig. 15. Moreover, considering the finite size of the supercell in the DFT calculations. It should be noted here that the dopant charge compensation was not considered in the DFT calculations. For the doped KNNs, the models were built by substituting K, Na, and Nb atoms with in-principle equivalent dopant atoms, i.e., Bi on the A-site, Sb and Zr on the B-site. For the vacancy-defect KNNs, the A-site atom was removed to produce K vacancy, while the corresponding oxygen vacancy was located along [001] and [010] directions in the orthorhombic $O_6$ octahedron. The total charges of the pure and doped KNNs were set to zero and the charge distribution was automatically balanced in the DFT calculation. The energy convergence criterion was set to be $10^{-6}$ eV and the structure was fully relaxed until the force was <0.002 eV/Å. We sampled the Brillouin zone integrations using $4 \times 4 \times 4$ Monkhorst-Pack grids for all the calculations.

**Dielectric and piezoelectric properties characterizations**. We measured the temperature-dependent relative permittivity and loss factor of KNN-based ceramics in a computer-controlled temperature chamber, which was connected to an LCR meter (E4980A, Agilent, Santa Clara, CA, USA). The samples were dried at 180 °C for 2 h before the test, and the measurement atmosphere was high-purity nitrogen. We measured the frequency-dependent loss factor of KNN- Bi,Sb,Zr ceramics by a dielectric/impedance analyzer (Novocontrol, Montabaur, Germany) under nitrogen atmosphere. We measured the room temperature and high-temperature piezoelectric coefficients using a quasi-static piezoelectric $d_{33}$-meter in a temperature chamber. We also measured the temperature-dependent piezoelectric coefficients by the resonance method according to the IEEE Standard on Piezoelectricity, using an HP4294 impedance analyzer[31]. We performed Rayleigh analysis and strain measurement in a piezo-measurement system (aixACCT TF Analyzer2000, aixACCT Systems GmbH, Aachen, Germany) with a high-voltage amplifier (TREK 610E, Medina, New York, USA).

**Microstructure observation by scanning transmission electron microscopy (STEM)**. Samples for electron microscopy were prepared by an FEI Helios NanoLab focused ion beam (FIB) by cutting the original polycrystalline ceramic sample into thin slices and then mounting the slices onto the Ominiprobe half grids. We performed STEM observation in a spherical aberration-corrected FEI Titan Themis G3 S/TEM equipped with a Schottky field-emission gun and operated at 300 kV. The beam current was 40 pA and had a convergence semi-angle of 17.8 mrad. We collected STEM images by a high-angle annular dark-field detector with inner- and outer-collection semi-angles of 73 and 200 mrad, respectively. We applied a drift correction mode to enable drift compensation by a cross-correlation algorithm. We collected 20 images with 1 K*1 K pixel size with a dwell time of 1 us/pixel and corrected using drift correction. We filtered noise in the collected STEM images by Gaussian filtering. Then, we conducted image analysis and quantification using custom MATLAB scripts. We found the atomic columns in A and B sites by searching for the local maximums in the STEM images. Then, we calculated the polar vector for each A-site atom by measuring the vector pointing from the local centre of mass of the surrounding B sites to the actual A site[30,38,43]. We determined the position of doped elements by comparing atomic column intensities in the HAADF-STEM images, in which the contrast is proportional to the atomic numbers of elements. For each atomic column (A or B site) found in the STEM images, we recorded its coordinates, intensity, and polar vector for further analyses, including drawing the intensity map, polar vector map, and polar vector angle map.

**Phase-field simulation**. We used phase-field simulations to show the impact of local structure on the properties and structure evolutions. In the phase-field model, we introduced the time-dependent Ginzburg-Landau (TDGL) equation to describe the temporal evolution of the polarization field and domain structure for the ferroelectric system with local structure[32], as follows:

$$\frac{\partial P_i(r,t)}{\partial t} = -L \frac{\delta F}{\delta P_i(r,t)}, \, i = (1, 2, 3) \qquad (3)$$

where $L$ is the kinetic coefficient, $F$ the total free energy of the system, **r** the space position, and $p_i$ (**r**, $t$) is the polarization. The total free energy of the system can be expressed as follows:

$$F = \int_V \left[ f_{bulk} + f_{elas} + f_{elec} + f_{grad} \right] dV \qquad (4)$$

where $V$ denotes the system volume, $f_{bulk}$ the Landau bulk free energy density, $f_{elas}$ the elastic energy density, $f_{elec}$ the electrostatic energy density and $f_{grad}$ the gradient energy density. The bulk free energy is expressed as follows:

$$\begin{aligned} f_{bulk} = &\, \alpha_1(P_1^2 + P_2^2 + P_3^2) + \alpha_{11}(P_1^4 + P_2^4 + P_3^4) + \alpha_{123}P_1^2P_2^2P_3^2 \\ &+ \alpha_{12}(P_1^2P_2^2 + P_1^2P_3^2 + P_2^2P_3^2) + \alpha_{111}(P_1^6 + P_2^6 + P_3^6) \\ &+ \alpha_{112}[P_1^4(P_2^2 + P_3^2) + P_2^4(P_1^2 + P_3^2) + P_3^4(P_1^2 + P_2^2)] \end{aligned} \qquad (5)$$

where $\alpha_1, \alpha_{11}, \alpha_{12}, \alpha_{111}, \alpha_{112}$ and $\alpha_{123}$ are Landau energy coefficients. The values of these coefficients determine the thermodynamic behaviours of the bulk phases.

Details for the rest of the contribution to the total free energy can be found in refs. [44,45].

We adopted a semi-implicit Fourier-spectral method for numerically solve the TDGL equation[46]. On the basis of the experimental data, we obtained the required Landau parameters[47]. In the phase-field model, the distinction should be bulk free energy that depends on the Landau coefficients between the tetragonal ferroelectric structure and the local structure. For the tetragonal ferroelectric structure, the Landau free energy parameters were as follows: $\alpha_1 = 2.05 \times 10^5 \times (T-550)$ $C^{-2}m^2N$, $\alpha_{11} = 2.13 \times 10^8$ $C^{-4}m^6N$, $\alpha_{12} = 7.78 \times 10^8$ $C^{-4}m^6N$, $\alpha_{111} = 1.26 \times 10^9$ $C^{-6}m^{10}N$, $\alpha_{112} = 5.76 \times 10^9$ $C^{-6}m^{10}N$, and $\alpha_{123} = -1.12 \times 10^{10}$ $C^{-6}m^{10}N$, where only a cubic-tetragonal phase transition could be found over the entire temperature range. For local structures, the parameters were as follows: $\alpha_1 = 2.1 \times 10^5 \times (T-620)$ $C^{-2}m^2N$, $\alpha_{11} = -3.57 \times 10^7$ $C^{-4}m^6N$, $\alpha_{12} = 1.06 \times 10^7$ $C^{-4}m^6N$, $\alpha_{111} = 5.52 \times 10^9$ $C^{-6}m^{10}N$, $\alpha_{112} = 2.26 \times 10^9$ $C^{-6}m^{10}N$ and $\alpha_{123} = 2.36 \times 10^{10}$ $C^{-6}m^{10}N$, where only a cubic-orthorhombic phase transition can be found over the entire temperature range, and T is the temperature in Kelvin. The elastic constants and electrostrictive coefficients are supposed to be the same in the whole tetragonal ferroelectric structure and local structures: $s_{11}^D = 5.6 \times 10^{-12}$ $m^2N^{-1}$, $s_{12}^D = -1.6 \times 10^{-12}$ $m^2N^{-1}$, $s_{44}^D = 13.1 \times 10^{-12}$ $m^2N^{-1}$, $Q_{11} = 0.166$, $Q_{12} = -0.072$, $Q_{44} = 0.042$[48–50]. In the computer simulations, we used two-dimensional $128 \times 128$ discrete grid points and periodic boundary conditions. The grid space in real space was chosen to be $\Delta x = \Delta y = 1$ nm.

## Data availability
The data that support the findings of this study are available from the corresponding authors upon reasonable request.

## Code availability
The custom computer codes used to generate the results reported in this article are available from the corresponding authors upon reasonable request.

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

## Acknowledgements

Q.S. and L.M.Z. acknowledge the support of the National Natural Science Foundation of China (51872217, 51932006, and 51521001) and the 111 Project (B13035). S.Z. acknowledge the support of ARC (FT140100698). F.L. acknowledge the support of the National Natural Science Foundation of China (51922083) and the 111 Project (B14040). J.S.W. acknowledge the support of the Fundamental Research Funds for Central Universities (WUT: 2019III012GX, 2019III190GX). H.L. and H.H. acknowledge the support of the Major Program of the Natural Science Foundation of China (51790490). X.Z.L. acknowledge the support of ARC (DP190101155). The authors are grateful for the scientific and technical assistance of the Microscopy Australia node at the University of Sydney (Sydney Microscopy and Microanalysis).

## Author contributions

The project was designed and conceived by X.Y.G., F.L., Z.B.C. and S.Z.; X.Y.G., Q.H.G., H.J.S., H.L., W.C. and H.H. performed the properties measurment; Z.X.C., J.L.W. and Q.F.G. performed the synchrotron XRD and data processing; X.Z. and B.L. performed the first-principle calculation; F.L. and Y.L. performed the phase-field simulations; Z.B.C., X.Y.M., X.Y.G., X.Z.L., S.P.R. and J.S.W. performed scanning transmission electron microscopy experiments and interpreted the results of scanning transmission electron microscopy experiments; Q.S., L.M.Z. and S.Z. supervised the experiments; X.Y.G., Z.B.C. and F.L wrote the manuscript, S.Z. revised the manuscript; All authors contributed to the discussion of the results.

## Competing interests

The authors declare no competing interests.
