## [Peer Review File · Nature Communications]

REVIEWER COMMENTS

Reviewer #1 (Remarks to the Author):

The manuscript by Gao et al. reports on the mechanism responsible for the enhanced piezoelectricity in multi-elements doped KNN-based ceramics. Authors have employed atomic-scale imaging, first-principle, and phase-field calculations, among other tools to identify the low-angle polar nano-domains as the critical factor in improving the dielectric and piezoelectric properties. There are some reports on the similar lines on lead-based perovskites (Adv. Funct. Mater. 2020, 2006823; Nature Materials 17 (2018), 349; Nature 546 (2017), 391, etc.) to explain the large piezoelectric response in relaxor ferroelectrics but such studies on KNN-based piezoelectric are lacking.

The results, explanations thereof presented in the manuscript are interesting and are worth publishing in Nature Communications. However, the following points must be addressed in the revised manuscript:

1. Reg. ceramic compositions: Authors have synthesized Sb, Bi, and Zr doped and co-doped samples. There is no mention of the formation of extrinsic defects in these samples. For instance, the substitution of 4% of K⁺ with Bi³⁺ in KNN would introduce additional oxygen in the structure (i.e., suppresses the oxygen vacancies or oxygen at interstitials type-defects if structure permits). Accordingly, the nominal composition should be K_{0.48}Bi_{0.02}Na_{0.5}Nb_{0.03}O₃. Zr⁴⁺ replacing Nb⁵⁺ would create oxygen vacancies (assuming A-site or B-site vacancy formation is not favorable for 1% Zr doped KNN). Similarly, for the multi-elements doped composition K_{0.48}Bi_{0.02}Na_{0.5}Nb_{0.92}Sb_{0.04}Zr_{0.04}O₃ has 0.02 [Bi]_K and 0.04 [Zr]_{Nb} defects.
2. What is the rationale behind choosing the Sb, Bi, and Zr as the dopants?
3. The relative density(ies) of the pellets used in this study should be explicitly stated. Figure S1 suggests significant porosity in some of these samples.
4. Line 368: What was the dwell time for the first stage of sintering (at 1190 °C) of KNN-Bi,Sb,Zr pellet?
5. P4mm and Amm2 should be changed P4mm and Amm2 (letters in S.G. should be formatted in italic.)
6. Page 5 and Figure S2: Rayleigh analysis: What is the typical coercive field (EC) for the KNN-Bi,Sb,Zr ceramic? Rayleigh law is valid for applied field E < EC. Is there any non-linearity in d₃₃ vs. E plot at E > 4 kV/cm?
7. Fig. S6: What was the peak shape function used for the Rietveld refinement of the XRD data? How was the FWHM of XRD peaks plotted in Fig. S4 and S6 calculated? (110)_{pc} is a doublet for Tetragonal crystal structure and a triplet for Orthorhombic crystal structure, whereas (111)_{pc} is a singlet in both Tetragonal and Orthorhombic crystal structures. So, is the value of FWHM (let us say at 300 K for peak (100) ~ 0.115 degrees for the entire broad peak (tetragonal + orthorhombic multiples) or individual peak due to reflection from one set of planes [e.g., Tetragonal (100)]).
8. Page 14 and Figure S6: What does the phrase "structural distortion" in the context of line 279 mean? A decrease in the degree of tetragonality (c/a ratio) is often labeled as a decrease in structural distortion. Further, in a system with tetragonal and orthorhombic phase coexistence, FWHM of (200)_{pc} peak could decrease if a) one of the phases disappears, or b) phase fractions remain constant, but the

degree of tetragonality or orthorhombicity decreases. Please elaborate.

9. Is there any noticeable dispersion in T_m (temperature of dielectric anomaly corresponding to orthorhombic to tetragonal transition) with frequency for KNN-Bi,Sb,Zr?

10. In the authors' opinion, what is the role of nature (size, charge valency, the activity of 6s² lone-pair, etc.) of the dopants as far the magnitude of the polar vector angle is concerned? Can we expect similar results if KNN-LS is co-doped with Sc, Ta, and Sn?

Reviewer #2 (Remarks to the Author):

Gao and co-workers have submitted a paper entitled "Origin of high piezoelectricity in multi-element doped KNN ceramics". In the recent couple of decades intense research efforts have been focused on the development of lead-free piezoelectric materials as alternatives to state-of-the art PZT materials. One of the most promising lead-free materials are KNN boosted by the seminal paper by Saito et al. published in Nature in 2004.

In this work unusually high piezoelectricity in multi-element doped KNN materials is reported for the first time, which only have been shown previously in highly textured KNN ceramics. In this work KNN ceramics with different levels of bB, Zr, Sb or Zr/Sb/bi doping were synthesized by conventional ceramic processing. The KNN ceramics were characterized by combination of temperature-dependent synchrotron x-ray diffraction, atomic-scale transmission electron microscopy, dielectric spectroscopy and measurement of piezoelectric coefficient, combined with first principles and phase field calculations. The unusually high piezoelectric coefficient was rationalised by a mechanism involving heterogeneity with respect to crystal symmetry and chemical composition. Nano-scale heterogeneity with different polar vectors were responsible for the high dielectric and piezoelectric responses.

The unusual high piezoelectric properties of these KNN ceramics is particularly interesting and the paper has therefore the potential to be published in Nature communication. Before publication the authors should address the following remarks to the manuscript.

The last paragraph before the Results chapter, contains results on both the microstructure, dielectric (including a Rayleigh analysis) and piezoelectric properties. These important data should be moved to the results chapter or is simply the heading (Results) displaced in the manuscript?

Before the authors start reporting on the properties of the KNN ceramics, the microstructure (supplementary Figure 1) and the density (Supplementary Table 1) of the materials needs to be reported in the first paragraph in the Results chapter. How was the density measured? The images for the single-doped samples gives an impression of a significantly higher porosity than inferred from the relative density. The authors should also provide SEM images of polished samples in order to give evidence of the low porosity/high density.

The multi-elements and single element doped samples were not prepared by the same sintering procedure. The mechanism presented is based on the relative comparison of the single and multi-elements doped sample, but the different processing procedure makes this questionable. How important is the sintering procedure for the properties for the multi-elements doped sample? The SEM image also give evidence for a very different microstructure of the multi-elements doped sample. (supplementary Figure 1). The work would be strengthened by also report on the single-element

doped samples sintered by the same procedure as the multi-elements doped sample.

The KNN precursor powders were prepared at 850 C and it can be questioned if all the dopants were dissolved in the KNN lattice prior to sintering. This would influence on the distribution of the dopants after sintering, particularly with respect to the two sintering procedures used.

The doping of KNN require also knowledge about the defect chemistry of the material, which also may influence on the electrical properties. The authors should consider the following points related to the defect chemistry caused by the doping:

- Zr has a lower formal charge than Nb and this difference need to be charge compensated for by an additional point defect(s), which one(s)? Correspondingly and more challenging, the Bi doping was obtained by simply substituting a K^+/Na^+ with Bi^{3+} . This is not possible without creating secondary phases or other charge compensation point defects. These considerations need to be commented in the manuscript and also discussed in relation to the heterogenous distribution of the dopants and the properties.
- How was the dopants charge compensated for in the DFT calculations and how did the authors distribute K and Na on the A-lattice in the supercells used? Would the distribution of K and Na affect the findings from DFT?
- How was loss of alkali-oxide during sintering compensated for or was this not considered (only nominal compositions are given)?

The co-existence of two phases in the multi-element doped sample point to chemical heterogeneity also evidenced by STEM. The authors should also provide X-ray diffraction data above T_c to demonstrate if this heterogeneity remains above T_c showing apparently two cubic KNN phase present. It would also be useful to report the unit cell parameters and volume as pseudo cubic values to make an easier comparison of the two KNN phases.

No ferroelectric properties were reported. It would be very useful for the quality of the paper to also report ferroelectric hysteresis loops.

In the evaluation of the STEM data the effect of the distribution of Bi and Zr were disregarded. Since the atomic number of Zr is so close to Nb this can be argued for, but the heavy element Bi with atomic number 83 cannot be disregarded. It is also confusing that no A-site and B-site intensity map is not reported for KNN-Bi,Sb,Zr. To make this analysis more convincing the same set of analysis are needed for both materials (KNN-Sb and KNN-Bi,Sb,Zr).

Doping has also recently been shown to affect the piezoelectric properties of PMN-PT single crystals, see Le et al. Science 364 (2019), 264. The authors should consider to make comparison with this work and similar works in other systems where minor doping have been shown to enhance piezoelectricity.

Reviewer #3 (Remarks to the Author):

The current contribution discusses the impact of a multiple of elements on the atomic level structural aspects and the piezoelectric properties of KNN ceramics. It is clearly noticed that the basic underlying idea is an extension from their earlier claims that unit cell level chemical heterogeneity that disturbs a long range polar order and consequently brings about an angular distribution in the direction of polar vector of each unit cell. I see that the collection of experimental data is of high quality, quite complete, and appropriate in supporting their claims, though I think I have to raise a coupling of questions to be clarified.

First, the presence of the claimed nano-heterogeneity is obvious from their extensive TEM studies, but a demonstration on how these nano-heterogeneity is contributing to the piezoelectricity seems missing. One could assume that the low-angle polar vectors contribute to the piezoelectricity when an electric field is applied through polarization rotation, but this is just a possibility. It could be that or could be a shear contribution from misoriented polar vectors or maybe something else. I do not see a decisive experimental verification on the claimed scenario here. In fact, their first principles calculation suggested that the coexistence of those multi-elements tends to reduce the anisotropy that is directly proportional to the magnitude of polar vectors. If this is so, the induced angular variation in the direction of polar vectors has a high chance not to make any useful contribution to the piezoelectricity. Please note that the piezoelectricity, especially the electromechanical strain, is a function of both the easy rotation and the magnitude of polar vectors. As noted, one trades off the other. In this sense, I think it would be better for the authors to tone down their claim a little by dropping the word "origin" in the title.

Second, the authors are consistent in emphasizing the effect of multi-elements to get the currently observed enhanced piezoelectricity by comparing the singly doped ones with the triply doped one. I agree that the latter leads to better enhanced piezoelectricity than the former. Then, how about introducing only two different elements? In fact, it has been well known that doubly introduced elements, Li and Sb, are also highly effective in enhancing the piezoelectricity. Is the underlying mechanism for the doubly doped KNN different from what is claimed in this work? How about more than three elements? Do the authors expect any room for further enhancement in piezoelectricity? In relation to this issue, I am also curious why the authors chose a mixture of A-site substituting Bi and B-site substituting Sb and Zr. Does the proposed mechanism only work with the choice of this type of mixture or also work with any combination of elements?

Responses to Referees' comments and the description of revisions in the revised manuscript and supplementary materials

We would sincerely thank the referees for their time and efforts in careful reading of the manuscript and in preparation of the review reports. We truly appreciate their positive comments, valuable questions and suggestions. We have revised our manuscript accordingly, we believed its quality had been greatly improved. The point-by-point responses to comments were enclosed in the following. We hope we have satisfactorily addressed all referees' concerns and questions. Also the revisions in the manuscript were highlighted.

Response to Reviewer #1

Comment 1: The manuscript by Gao et al. reports on the mechanism responsible for the enhanced piezoelectricity in multi-elements doped KNN-based ceramics. Authors have employed atomic-scale imaging, first-principle, and phase-field calculations, among other tools to identify the low-angle polar nano-domains as the critical factor in improving the dielectric and piezoelectric properties. There are some reports on the similar lines on lead-based perovskites (Adv. Funct. Mater. 2020, 2006823; Nature Materials 17 (2018), 349; Nature 546 (2017), 391, etc.) to explain the large piezoelectric response in relaxor ferroelectrics but such studies on KNN-based piezoelectric are lacking. The results, explanations there of presented in the manuscript are interesting and are worth publishing in Nature Communications. However, the following points must be addressed in the revised manuscript:

Reply: We thank the reviewer for his/her positive and valuable comments. We also thank the reviewer for providing references with similar topic on lead-based perovskite, we added the reference in our revised paper and discussed the mechanism explaining the large piezoelectric response in different relaxor ferroelectric systems. In addition, we will respond to the comments one by one in the following.

Comment 2: Reg. ceramic compositions: Authors have synthesized Sb, Bi, and Zr doped and co-doped samples. There is no mention of the formation of extrinsic defects in these samples. For instance, the substitution of 4% of K^+ with Bi^{3+} in KNN would introduce additional oxygen in the structure (i.e., suppresses the oxygen vacancies or oxygen at interstitials type-defects if structure permits). Accordingly, the nominal composition should be $K_{0.48}Bi_{0.02}Na_{0.5}NbO_{3.02}$. Zr^{4+} replacing Nb^{5+} would create oxygen vacancies (assuming A-site or B-site vacancy formation is not favorable for 1% Zr doped KNN). Similarly, for the multi-elements doped composition $K_{0.48}Bi_{0.02}Na_{0.5}Nb_{0.92}Sb_{0.04}Zr_{0.04}O_3$ has $0.02 Bi_{K/Na}$ and $0.04 Zr'_{Nb}$ defects.

Reply: We truly appreciate the valuable comments. We totally agreed with the reviewer that the dopants can induce extrinsic defects, which play important roles in impacting the macroscopic properties. As the reviewer pointed out, A-sites and oxygen vacancies will be introduced by donor dopant on A- site and acceptor dopant on B-site of KNN, respectively. We performed the first-principle calculations based on density functional theory to optimize the structure of A-site vacancies and oxygen vacancies to demonstrate the contributions of the defects on microstructure, as show in the following Fig. R1 (Supplementary Fig. 7 in the revised manuscript).

Fig. R1. The contributions of the A-site and oxygen vacancies to the structure. The six B-O bonds of KNNs after adding oxygen vacancies approach the equal length, while the A-site vacancies including Na-vacancy and K-vacancy have much weaker effect on the length of the B-O bonds. (Supplementary Fig. 7 in the revised manuscript)

We added the following paragraphs in the main text to discuss the contributions of the defects.

“It should be noted that the additions of Bi^{3+} and Zr^{4+} to A- and B- sites of KNN ceramics will inevitably introduce vacancy defects such as A-site vacancies and oxygen vacancies, respectively. The contributions of the A-site and oxygen vacancies to the structure were also studied based on first-principle calculations and given in Supplementary Fig. 7. Interestingly, the difference in the six B-O bond lengths of the octahedron, after adding different dopants (Fig. 1e) and oxygen vacancies, is obviously reduced compared with the undoped KNN. On the contrary, the A-site vacancies including Na-vacancy and K-vacancy have much weaker effect on the length of the B-O bonds.”

“It should be noted here that dopants generally induce vacancies, for example, the Bi^{3+} occupying A-site K^+/Na^+ position will cause A-site vacancies (including V_{K}' or V_{Na}'), while the Zr^{4+} occupying B-site Nb^{5+} position will lead to oxygen vacancies (V_{O}), which are expected to impact the dielectric and piezoelectric properties. In our studied multi-elements doped KNN, we attempted to keep the charge balanced by tuning the composition with the appropriate doped levels, thus the effect of defects induced by dopants is minimal and is not considered in our mechanism.”

Comment 3: What is the rationale behind choosing the Sb, Bi, and Zr as the dopants?

Reply: Thanks for the good question. The piezoelectricity of KNN was reported to be effectively enhanced by multi-elements dopant and ascribed to the construction of room temperature rhombohedral-tetragonal morphotropic phase boundary [*Annu. Rev. Mater. Res.* 48, 191-217 (2018)]. Unfortunately we cannot detect the rhombohedral phase in any of the systems, therefore the motivation of our present work is to explore the real underlying mechanism responsible for the high piezoelectricity in multi-elements doped KNN, we chose the Sb, Bi and Zr doped KNN as an example

system. We performed synchrotron X-ray diffractions, TEM experiments and first-principle calculations on several multi-elements doped KNN systems, try to understand the relationship between the macroscopic properties and microstructure.

Comment 4: The relative density(ies) of the pellets used in this study should be explicitly stated. Figure S1 suggests significant porosity in some of these samples.

Reply: We appreciate the valuable comment. We have added the following statements in the method part to explain the way we measured the relative density:

“The densities of the ceramics were measured by the Archimedes method, where the relative density was calculated based on the theoretical density of KNN.³⁹”

We are sorry about the low quality SEM images given in Supplementary Fig. 1 in previous manuscript, where we used the non-polished samples. In order to clearly show the microstructure and grain size, new SEM images on polished and thermal etched samples were given in the revised manuscript, as shown in the Fig. R2 (Supplementary Fig. 1 in the revised manuscript).

It should be noted here that the fissures between different grains in Figs. R2 (b) & (d), were associated with the high etching temperature, other than the voids or pores in the samples. Here we also gave the SEM images before the thermal etching treatments, as shown in Fig. R3, exhibiting a very low porosity. Many small pores were observed in KNN-Bi,Sb,Zr (Fig. R3a), and was associated with the abnormal grain growth [*J. Mater. Chem. C*, 8, 7606-7649 (2020)].

Fig. R2. The FE-SEM images of multi-elements and single-element doped KNN ceramics. (a) The FE-SEM image for multi-elements doped KNN ceramic (KNN-Bi,Sb,Zr). (b-d) The FE-SEM images for KNN-Bi, KNN-Sb, KNN-Zr ceramics, respectively. (Supplementary Fig. 1 in the revised manuscript)

Fig. R3. The FE-SEM images of multi-elements and single-element doped KNN ceramics before the thermal etching. (a) The FE-SEM image for multi-elements doped KNN ceramic (KNN-Bi,Sb,Zr). (b-d) The FE-SEM images for KNN-Bi, KNN-Sb, KNN-Zr ceramics, respectively.

Comment 5: Line 368: What was the dwell time for the first stage of sintering (at 1190 °C) of KNN-Bi,Sb,Zr pellet?

Reply: Thanks for the question. For the sintering process, there is no holding time at 1190 °C. The samples were immediately cooled down to 1090°C at a rate of 10 °C/min. The schematic diagram of the sintering process is given in the following Fig. R4.

Fig. R4. The schematic diagram of the sintering process.

Comment 6: P4mm and Amm2 should be changed *P4mm* and *Amm2* (letters in S.G. should be formatted in italic.)

Reply: Thanks for the careful reading and sorry about the mistake. The P4mm and Amm2 have been changed to the *P4mm* and *Amm2*.

Comment 7: Page 5 and Figure S2: Rayleigh analysis: What is the typical coercive field (EC) for the KNN-Bi,Sb,Zr ceramic? Rayleigh law is valid for applied field $E <$

EC. Is there any non-linearity in d_{33} vs. E plot at $E > 4$ kV/cm?

Reply: Thanks for the valuable comment and question. The ferroelectric hysteresis loop has been added in the supporting information, as shown in Fig. R5 (Updated Supplementary Fig. 2 in the revised manuscript). The value of E_c is about 8 kV/cm, the applied field in our Rayleigh study was below the half of the coercive field.

Fig. R5 Rayleigh study and the intrinsic/extrinsic contributions. (a) The ferroelectric hysteresis loop of KNN-Bi,Sb,Zr ceramic. (b) Comparison between the measured and calculated strain-versus-electric field hysteresis loop of KNN-Bi,Sb,Zr ceramic. (c) The ac electric field-dependent piezoelectric coefficient d_{33} and Rayleigh parameters for KNN-based ceramics. (d) The ratio of extrinsic contribution $\alpha E_0 / (\alpha E_0 + d_{init})$ for KNN-based ceramics. (Supplementary Fig. 2 in the revised manuscript).

Fig. R6. The d_{33}^* as a function of applied electric field.

Fig. R6 gives d_{33}^* as a function of the applied electric field. The d_{33}^* values were found to increase first and then decrease with increasing the electric field. Above 4

kV/cm, the d_{33}^* value goes down above 10kV/cm because higher electric field will clamp the domain wall motion and lead to the decreased piezoelectricity, showing a high non-linearity.

Comment 8: Fig. S6: What was the peak shape function used for the Rietveld refinement of the XRD data? How was the FWHM of XRD peaks plotted in Fig. S4 and S6 calculated? (110)pc is a doublet for Tetragonal crystal structure and a triplet for Orthorhombic crystal structure, whereas (111)pc is a singlet in both Tetragonal and Orthorhombic crystal structures. So, is the value of FWHM (let us say at 300 K for peak (100) ~ 0.115 degrees for the entire broad peak (tetragonal + orthorhombic multiples) or individual peak due to reflection from one set of planes [e.g., Tetragonal (100)].

Reply: Thanks to the reviewer for the valuable comments. We used the pseudo-Voigt convoluted with axial divergence asymmetry function for the Rietveld refinement of the XRD data. The FWHM of XRD peaks plotted in Supplementary Figs. 4, and 6 were derived from the peak analysis using Origin software built-in package. The values of FWHM shown in Supplementary Figs. 4, and 6 stand for the entire broad peak (tetragonal+orthorhombic). This gives us an indication of how the crystal structure changes with temperature and where the transition temperature may locate. To make this clear, we added the following sentence in the method of Structure characterization by synchrotron radiation X-ray diffraction (synchrotron XRD) section:

“The values of full width at half maximum stand for the entire broad peak, derived from peak analysis using the Origin software built-in package.”

Comments 9: Page 14 and Figure S6: What does the phrase “structural distortion” in the context of line 279 mean? A decrease in the degree of tetragonality (c/a ratio) is often labeled as a decrease in structural distortion. Further, in a system with tetragonal and orthorhombic phase coexistence, FWHM of (200)pc peak could decrease if a) one of the phases disappears, or b) phase fractions remain constant, but the degree of tetragonality or orthorhombicity decreases. Please elaborate.

Reply: Thanks for the valuable comments. The structural distortion here represents how far the unit cell deviates from Cubic unit cell. We agree with the reviewer that the c/a ratio closer to 1 (e.g. the decrease in the degree of tetragonality (c/a ratio)) means a decrease in structural distortion. We also agree that the decrease in FWHM of (200)pc peak could have other possible sources in a system with the coexistence of tetragonal and orthorhombic phases. Here, based on our refinement of XRD for KNN-Bi,Sb,Zr as a function of temperature, we can conclude that the decrease of the FWHM for (200)pc peak in Supplementary Fig. 6 is due to the disappearance of orthorhombic phase (please see Fig. 1) with increasing temperature. By comparing different compositions, the decreased FWHM of (200)pc means the degree of tetragonality or orthorhombicity decreases. We deleted “structural distortion” from the results discussion for a clear expression, we also changed the caption of Supplementary Fig. 6, as shown in the following:

“It is interesting to note that the FWHM values of KNN-Bi,Sb,Zr sample are lower than those of KNN-Sb counterpart (Supplementary Fig. 4), indicating the tetragonality or orthorhombicity of KNN-Bi,Sb,Zr sample is decreased comparing to the KNN-Sb sample.”

Comments 10: Is there any noticeable dispersion in T_m (temperature of dielectric anomaly corresponding to orthorhombic to tetragonal transition) with frequency for KNN-Bi,Sb,Zr?

Reply: Thanks to the good question. Yes, we observed noticeable dispersion around the temperature of dielectric anomaly corresponding to orthorhombic to tetragonal transition.

To make this clear, we added the Fig. R7 (Supplementary Fig. 11 in the revised manuscript) in the supporting information.

Fig. R7 The dielectric loss factor and relative permittivity as a function of frequency for KNN-Bi,Sb,Zr ceramic over temperature range of 153–293 K. (a-b) dielectric loss factor, (c-d) relative permittivity. (Supplementary Fig. 11 in the revised manuscript)

Comments 11: In the authors' opinion, what is the role of nature (size, charge valency, the activity of 6s2 lone-pair, etc.) of the dopants as far the magnitude of the polar vector angle is concerned?

Reply: We appreciate the valuable question. In our opinion, the magnitude of the polar vector angle mainly corresponds to the dopant element distribution and concentration, where the nature of the dopants plays important roles. This can be confirmed by the first principle calculation, where the nature of the dopants, including the size and charge valence, are already considered during the calculations, and we can see that the dopants will cause the B-O bonds of the octahedrons approaching the same length, as shown in Figs. 1e and R1.

Based on the current techniques, however, it is hard to distinguish the impact of size/charge on the polar vector which might require three dimensional structure evolution on atomic scale. In our STEM observation instead, we used the 2D mapping of the polar vector angle evolution considering the dopant distribution and

concentration.

Comments 12: Can we expect similar results if KNN-LS is co-doped with Sc, Ta, and Sn?

Reply: We appreciate the valuable question. Based on the references and our recent works, we thought the KNN-LS with the addition of Sc/Bi (we need adding the Bi³⁺ on A-site to keep charge neutrality if we dope Sc³⁺ on B-site) had minor contribution to the piezoelectricity while the addition of Ta⁵⁺ would greatly benefit the piezoelectricity enhancement. Meanwhile, very minor Sn⁴⁺ addition will increase the piezoelectricity because of enhanced local structure heterogeneity, however large amount of Sn⁴⁺ dopant will decrease the piezoelectricity since Sn⁴⁺ with full d¹⁰ electronic configuration will not favor the ferroelectricity based on the pseudo Jahn-Teller effect.

The detailed discussions are shown in below:

- (1) Effect of Ta⁵⁺. There were many publications reporting the similar multi-elements doped KNN ceramics with desired piezoelectricity. The d_{33} value of Li/Sb doped KNN is about 260 pC/N [*J. Appl. Phys.*, 100, 104108, (2006)], increasing to 400 pC/N with Ta dopant in addition to Li/Sb [*J. Am. Ceram. Soc.*, 92, 283-285 (2009)].
- (2) Effect of Sc³⁺. We also fabricated Li, Sb, Sc, Bi doped KNN recently. The piezoelectric of multi-elements doped KNN is about 300 pC/N at room temperature. The valence of Sc³⁺ is lower than Nb⁵⁺; thus, the Bi³⁺ doping element was chosen replacing A-site cations to balance the charge. The temperature dependent dielectric property is shown in Fig. R8 (a).
- (3) Effect of Sn⁴⁺. The Li, Sb, Sc, Bi, Sn doped KNN was also prepared, with d_{33} value of 340 pC/N. The temperature dependent dielectric property is shown in Fig. R8 (b).
- (4) Of interest is that the d_{33} value of Li, Sb, Sc, Bi, Ta doped KNN is above 400 pC/N, with temperature dependent dielectric property given in Fig. R8 (c).

Fig. R8. The temperature dependence of relative permittivity and loss factor for (a) KNN-Li,Sb,Bi,Sc, (b) KNN-Li,Sb,Bi,Sc,Sn, and (c) KNN-Li,Sb,Bi,Sc,Ta, respectively.

Response to Reviewer #2

Comment 1: Gao and co-workers have submitted a paper entitled “Origin of high piezoelectricity in multi-element doped KNN ceramics”. In the recent couple of decades intense research efforts have been focused on the development of lead-free piezoelectric materials as alternatives to state-of-the art PZT materials. One of the most promising lead-free materials are KNN boosted by the seminal paper by Saito et al. published in Nature in 2004. In this work unusually high piezoelectricity in multi-element doped KNN materials is reported for the first time, which only have been shown previously in highly textured KNN ceramics. In this work KNN ceramics with different levels of Bi, Zr, Sb or Zr/Sb/bi doping were synthesized by conventional ceramic processing. The KNN ceramics were characterized by combination of temperature-dependent synchrotron x-ray diffraction, atomic-scale transmission electron microscopy, dielectric spectroscopy and measurement of piezoelectric coefficient, combined with first principles and phase field calculations. The unusually high piezoelectric coefficient was rationalised by a mechanism involving heterogeneity with respect to crystal symmetry and chemical composition. Nano-scale heterogeneity with different polar vectors were responsible for the high dielectric and piezoelectric responses. The unusual high piezoelectric properties of these KNN ceramics is particularly interesting and the paper has therefore the potential to be published in Nature communication. Before publication the authors should address the following remarks to the manuscript.

Reply: We thank the reviewer for the positive comments and valuable suggestions.

Comment 2: The last paragraph before the Results chapter, contains results on both the microstructure, dielectric (including a Rayleigh analysis) and piezoelectric properties. These important data should be moved to the results chapter or is simply the heading (Results) displaced in the manuscript?

Reply: Thanks for the valuable suggestion. We have moved this paragraph to the result part according to the suggestion and highlight the change in the main text.

Comment 3: Before the authors start reporting on the properties of the KNN ceramics, the microstructure (supplementary Figure 1) and the density (Supplementary Table 1) of the materials needs to be reported in the first paragraph in the Results chapter. How was the density measured? The images for the single-doped samples gives an impression of a significantly higher porosity than inferred from the relative density. The authors should also provide SEM images of polished samples in order to give evidence of the low porosity/high density.

Reply: We appreciate the valuable comments. We added the following sentences in the first paragraph in the results chapter, reporting the microstructure and density, we also added the density measurement details in the section of the method.

“The multi-elements doped KNN is found to possess much larger grain size, being on the order of 25 μ m, one order larger than the single element doped counterparts, though all of the samples have a similar density of 94-96%.”

“The densities of the ceramics were measured by the Archimedes method, where the relative density was calculated based on the theoretical density of KNN.³⁹”

In order to clearly show the microstructure and grain size, the SEM images of the polished and thermally etched samples were given in the revised manuscript, as

shown in Fig. R9 (Supplementary Fig. 1 in the revised manuscript).

It should be noted here that the fissures between different grains in Figs. R9 (b) and (d), were associated with the high etching temperature, other than the voids or pores in the samples. Here we also gave the SEM images before the thermal etching treatments, as in Fig. R10, showing a very low porosity/high density. Many small pores were observed in KNN-Bi,Sb,Zr (Fig. R10a), this was associated with the abnormal grain growth [J. Mater. Chem. C, 8, 7606-7649 (2020)].

Fig. R9 The FE-SEM images of multi-elements and single-element doped KNN ceramics. (a) The FE-SEM image for multi-elements doped KNN ceramic (KNN-Bi,Sb,Zr). (b-d) The FE-SEM images for KNN-Bi, KNN-Sb, KNN-Zr ceramics, respectively. (Supplementary Fig. 1 in the revised manuscript)

Fig. R10. The FE-SEM images of multi-elements and single-element doped KNN ceramics before the thermal etching. (a) The FE-SEM image for multi-elements doped KNN ceramic (KNN-Bi,Sb,Zr). (b-d) The FE-SEM images for KNN-Bi, KNN-Sb, KNN-Zr ceramics, respectively.

Comment 4: The multi-elements and single element doped samples were not

prepared by the same sintering procedure. The mechanism presented is based on the relative comparison of the single and multi-elements doped sample, but the different processing procedure makes this questionable. How important is the sintering procedure for the properties for the multi-elements doped sample? The SEM image also give evidence for a very different microstructure of the multi-elements doped sample. (supplementary Figure 1). The work would be strengthened by also report on the single-element doped samples sintered by the same procedure as the multi-elements doped sample.

The KNN precursor powders were prepared at 850 C and it can be questioned if all the dopants were dissolved in the KNN lattice prior to sintering. This would influence on the distribution of the dopants after sintering, particularly with respect to the two sintering procedures used.

Reply: We appreciate the valuable comments and questions. The multi-elements and single element doped samples were not prepared by the same sintering procedure, since different dopants doped KNN ceramics were sintered according to their optimized sintering conditions. This is because the sintering temperature window of KNN-based ceramic is very narrow [*J. Am. Ceram. Soc.* 94, 3659-3665, (2011)]. Following reviewer's suggestion, we tried to use the same procedure to prepare different elements doped KNN ceramics, as shown in Fig. R11. The single element doped KNNs were not sintered or over-sintered, the samples had very high dielectric loss and could not be poled completely, with piezoelectric coefficients below 50 pC/N.

Fig. R11. The pictures of KNN-Bi, KNN-Sb, and KNN-Zr ceramics.

We checked the precursor powders of our studied compositions by XRD. According to the results shown in Fig. R12, the precursor powders did not show any impurity phase or secondary phase, demonstrating that all the dopants were dissolved in the KNN lattice.

Fig. R12. The XRD patterns of calcined powders of KNN-Bi,Sb,Zr, KNN-Zr, KNN-Sb, and KNN-Bi.

We totally agree with the reviewer that different microstructures exist in the multi-elements and single element doped KNNs. In order to study the role of different microstructures impacting the piezoelectric properties, we performed TEM to study the domain structures of KNN-Bi,Sb,Zr and comparing to KNN-Sb, as shown in the Fig. R13 (Supplementary Fig. 14 in the revised manuscript).

Fig. R13. The domain structure images of KNN-Sb and KNN-Bi,Sb,Zr ceramics. (a) The domain structure of KNN-Sb ceramic. The domain size is about 100 nm in the small grain. (b) The domain structure of KNN-Bi,Sb,Zr ceramic. The domain is about 500 nm in large grain, which coupled with the hierarchical domain structure (mark with the white circle). (Supplementary Fig. 14 in the revised manuscript)

As shown in Fig. R13, the grain size of KNN- Bi,Sb,Zr is much bigger than that of KNN-Sb. The domain size is proportional to the square root of grain size, thus the domain size of KNN- Bi,Sb,Zr was found to be about 500 nm, five time that of KNN-Sb (~100 nm). Of particular importance is that the hierarchical domain structure was observed in the large laminar domain of KNN-Bi,Sb,Zr ceramics, accounting for the greatly enhanced piezoelectric properties [*Natl. Sci. Rev.* 7, 355-365 (2020)].

Based on the above discussion, we also added the domain structure observation in the revised manuscript, as shown in the following.

“In addition, the high mobility of the 90° domain walls and hierarchical domain structure (Supplementary Fig. 14) in the large grain of KNN-Bi,Sb,Zr ceramics may also be an important contributor to the enhanced extrinsic piezoelectricity.^{22,33,34,37}”

Comment 5: The doping of KNN require also knowledge about the defect chemistry of the material, which also may influence on the electrical properties. The authors should consider the following points related to the defect chemistry caused by the doping: Zr has a lower formal charge than Nb and this difference need to be charge compensated for by an additional point defect(s), which one(s)? Correspondingly and more challenging, the Bi doping was obtained by simply substituting a K^+/Na^+ with Bi^{3+} . This is not possible without creating secondary phases or other charge compensation point defects. These considerations need to be commented in the manuscript and also discussed in relation to the heterogenous distribution of the dopants and the properties.

Reply: We truly appreciate the valuable comments. We agree that the defects induced

by the dopants are very important impacting the microstructure and properties. In this system, the Zr^{4+} has lower valence thus generating oxygen vacancy when replacing Nb^{5+} . While the Bi^{3+} was used to provide the charge compensation on A-site by A-site vacancies because the Bi^{3+} has higher valence comparing to Na^+/K^+ . We performed the first-principle calculations based on density functional theory to optimize the structure of A-site vacancies and oxygen vacancies, tried to understand the impact of oxygen vacancy and Na/K vacancies on the phase structure, as shown in Fig. R14 (Supplementary Fig. 7 in the revised manuscript).

Fig. R14 The contributions of the A-site and oxygen vacancies to the structure. The six B-O bonds of KNNs after adding oxygen vacancies approach the equal length, while the A-site vacancies including Na-vacancy and K-vacancy have much weaker effect on the length of the B-O bonds. (Supplementary Fig. 7 in the revised manuscript)

Based on reviewer's suggestion, we added the following paragraphs in the main text to describe the contribution of the defects.

"It should be noted that the additions of Bi^{3+} and Zr^{4+} to A- and B- sites of KNN ceramics will inevitably introduce vacancy defects such as A-site vacancies and oxygen vacancies, respectively. The contributions of the A-site and oxygen vacancies to the structure were also studied based on first-principle calculations and given in Supplementary Fig. 7. Interestingly, the difference in the six B-O bond lengths of the octahedron, after adding different dopants (Fig. 1e) and oxygen vacancies, is obviously reduced compared with the undoped KNN. On the contrary, the A-site vacancies including Na-vacancy and K-vacancy have much weaker effect on the length of the B-O bonds."

"It should be noted here that dopants generally induce vacancies, for example, the Bi^{3+} occupying A-site K^+/Na^+ position will cause A-site vacancies (including V'_K or V'_{Na}), while the Zr^{4+} occupying B-site Nb^{5+} position will lead to oxygen vacancies ($V_{\dot{O}}$), which are expected to impact the dielectric and piezoelectric properties. In our studied multi-elements doped KNN, we attempted to keep the charge balanced by tuning the composition with the appropriate doped levels, thus the effect of defects induced by dopants is minimal and is not considered in our mechanism."

Comment 6: How was the dopants charge compensated for in the DFT calculations and how did the authors distribute K and Na on the A-lattice in the supercells used? Would the distribution of K and Na affect the findings from DFT? How was loss of alkali-oxide during sintering compensated for or was this not considered (only nominal compositions are given)?

Reply: We appreciate the valuable questions and comments. For doped KNNs, the A-site or O vacancies may exist due to charge compensation and the volatile nature of K/Na atoms. However, considering the finite size of the supercell in the DFT calculations (a $2 \times 2 \times 2$ 40 atoms supercell that has been used to improve the computational efficiency while still maintaining the local structural characteristics), it is improper to incorporate one dopant atom and one A-site or O vacancy together in such a cell, as this would introduce an unrealistic high vacancy defect concentration. Besides, the effect of the vacancy defects induced by doping in experiments is regarded as minimal by carefully tuning the composition with appropriate doped levels. In view of these, the dopant charge compensation was not considered explicitly with the charge balanced automatically in the calculations, and more attention has been paid to understand the local structural properties of the regions around the dopants, i.e., the lengths of the six B-O bonds.

We also added the following sentence and Fig. R16 (Supplementary Fig. 15 in the revised manuscript) in the method and supporting information to describe the distribution of K and Na on the A-lattice in the supercells.

“The supercell with the uniform distribution of A-site cations along the *a*, *b*, and *c* directions was used in the calculations, as shown in the supplementary Fig. 15.”

Fig. R16 The supercell of pure KNN that used to perform first-principles calculations. The purple, yellow, green, and red atoms are K, Na, Nb, and O atoms, respectively. The supercell with the uniform distribution of A-site cations along the *a*, *b*, and *c* directions was used in the calculations (Supplementary Fig. 15 in the revised manuscript).

We added the following sentence to describe the method used to perform the DFT calculations for doped KNN:

“For the doped KNNs, the models were built by substituting K, Na, and Nb atoms with in-principle equivalent dopant atoms, i.e., Bi on the A site, Sb and Zr on the B-site.”

Based on the comments, we performed the DFT calculations with another distribution of A-site cations supercell (The supercell of nonuniform distribution as shown in Fig. R17a), as shown in the Fig. R17b. The results shown that the nonuniform distribution supercell had slightly influence on the length of B-O bonds comparing to the supercell of uniform distributed K and Na elements. But the evolution of B-O bonds length of nonuniform distribution supercell, after adding different dopants, was similar with the uniform distribution supercell results. In fact, after adding different dopants, the difference of B-O bonds length of nonuniform and uniform distribution supercell was all obviously reduced compared with the undoped KNN. In view of these, the distribution of K and Na elements in DFT calculations may have small effect for the trend, that the difference of six B-O bond lengths of the octahedron, after adding different dopants, is obviously reduced compared with the undoped KNN.

Fig. R17. (a) The supercell of nonuniform A-sites distribution of pure KNN that used to perform first-principles calculations. (b) The lengths of the six B-O bonds of perovskite octahedron for undoped and single element doped KNNs obtained from first-principles calculations of nonuniform A-sites distribution supercell.

We didn't consider the loss of alkali-oxide during sintering, and only used the nominal composition in the article. In fact, the loss of alkali-oxide during sintering will lead to A-site vacancies, including K-vacancy and Na-vacancy, have very weak impact on the B-O bond lengths based on first-principles calculations, as discussed above.

Comment 7: The co-existence of two phases in the multi-element doped sample point to chemical heterogeneity also evidenced by STEM. The authors should also provide X-ray diffraction data above T_c to demonstrate if this heterogeneity remains above T_c showing apparently two cubic KNN phase present. It would also be useful to report the unit cell parameters and volume as pseudo cubic values to make an easier comparison of the two KNN phases.

Reply: We thank the reviewer for the valuable suggestions. Based on the comments, we performed in-situ XRD as a function of temperature, and did the refinement to exhibit the phase structure evolution at high temperature (the X-ray diffraction data near and above T_c). We used two models ($P4mm$ and $Pm-3m$) for the refinement. The database CIF files were ICSD 173741 and ICSD 186364 for $P4mm$ and $Pm-3m$, respectively. The FWHM was found to slowly decrease at temperature above T_c . The refinement results shown that the Tetragonal and Cubic phases coexisted over

temperature range from 500 K to 700 K, above which, the pure Cubic phase could be observed.

The XRD data were given in Fig. R18 (Supplementary Fig. 10 in the revised manuscript).

Fig. R18 The X-ray diffraction results for KNN-Bi,Sb,Zr ceramic over temperature range of 500–800 K. (a) The X-ray diffraction patterns for KNN-Bi,Sb,Zr ceramic. (b) The full width at half maximum (FWHM) values of (110) and (100) peaks. (c) The phase fraction and unit cell volume of the tetragonal and Cubic phases. The temperature of phase transition from tetragonal to Cubic phase is around 550 K, while the tetragonal phase yet exists over temperature above the T_c . (d) The lattice parameters of tetragonal and Cubic phases. (Supplementary Fig. 10 in the revised manuscript)

We revised the following paragraph in the main text to describe the evolution of phase structure at temperature above T_c .

“Fig. 2a gives dielectric properties of KNN-Bi,Sb,Zr ceramics over a temperature range of 130–700 K, with dielectric anomalies at 550 K and 290 K, which correspond to Curie temperature and O–T phase transition temperature (Fig. 1d), respectively. It should be noted that the tetragonal phase still exists above the Curie temperature over a broad temperature range even the volume fraction is well below 10% (Supplementary Fig. 10), which can also be confirmed by the broad dielectric peak at 550 K as shown in Fig. 2a.”

Comment 8: No ferroelectric properties were reported. It would be very useful for the quality of the paper to also report ferroelectric hysteresis loops.

Reply: We thank the reviewer for the good suggestion. The ferroelectric hysteresis loop was added in Fig. R19 (Supplementary Fig. 2 in the revised manuscript). In addition, the coercive fields of KNN-Sb, KNN-Zr, KNN-Bi, were found to be on the order of 11, 13, and 18 kV/cm, respectively, and added in the modified Supplemental Table 1.

Fig. R19 Rayleigh study and the intrinsic/extrinsic contributions. (a) The ferroelectric hysteresis loop of KNN-Bi,Sb,Zr ceramic. (b) Comparison between the measured and calculated strain-versus-electric field hysteresis loop of KNN-Bi,Sb,Zr ceramic. (c) The ac electric field-dependent piezoelectric coefficient d_{33} and Rayleigh parameters for KNN-based ceramics. (d) The ratio of extrinsic contribution $\alpha E_0 / (\alpha E_0 + d_{int})$ for KNN-based ceramics. (Supplementary Fig. 2 in the revised manuscript).

Comment 9 In the evaluation of the STEM data the effect of the distribution of Bi and Zr were disregarded. Since the atomic number of Zr is so close to Nb this can be argued for, but the heavy element Bi with atomic number 83 cannot be disregarded. It is also confusing that no A-site and B-site intensity map is not reported for KNN-Bi,Sb,Zr. To make this analysis more convincing the same set of analysis are needed for both materials (KNN-Sb and KNN-Bi,Sb,Zr).

Reply: We thank the reviewer for the valuable comments. As the reviewer pointed out, it is hard to identify Zr on the B-site of KNN because the atomic number of Zr is so close to Nb. We also did the A-sublattice intensity analysis, as given in Fig. R20, unfortunately, it is difficult for us to separate the Bi dopant rich areas, even the atomic number of Bi is much larger than both K and Na elements, due to the following reasons:

In the Sb doped KNN sample, K and Na are not randomly distributed. Local K and Na segregations were observed in Fig. R20 (a), where the high intensity (bright) areas belong to K rich regions because K has atomic number 19, being larger than that of Na (atom number 11). As the Bi was added, high contrast areas were also observed, as shown in Fig. R20 (b). However, it is difficult to identify the actual element that contributes to the observed STEM-HAADF contrast. With the coexistence of Bi, K and Na elements, the brighter contrast areas can be either K-rich or Bi-rich areas that cannot be well defined. In addition, the amount of Bi-dopants is very small, which might not contribute to the contrast of the STEM-HAADF image too much considering the large amount of K element. Therefore, we could not give a solid proof of the Bi rich area from STEM-HAADF images.

Fig. R20. Normalized intensity of the A-site for the (a) KNN-Sb and (b) KNN-Bi,Sb,Zr ceramics

Comment 10: Doping has also recently been shown to affect the piezoelectric properties of PMN-PT single crystals, see Le et al. *Science* 364 (2019), 264. The authors should consider to make comparison with this work and similar works in other systems where minor doping have been shown to enhance piezoelectricity.

Reply: Thanks for the valuable suggestion. We added recently research on lead-based relaxor ferroelectrics in our paper and make comparisons accordingly [*Science* 364, 264-268, (2019); Kumar, A., et al. *Nat. Mater.* (2020); and *Adv. Funct. Mater.* 2006823, (2020)]. Please check the following paragraph.

“Interestingly, the rare-earth doped PMN-PT ceramics with enhanced piezoelectric coefficient also exhibit increased volume fraction of tetragonal phase with reduced Curie temperature and broad relaxation peak.^{31,36} Tetragonal phase with low-angle domain wall was also observed in the structure evolution of PMN-PT solid solution with increasing Ti concentration based on the molecular dynamic calculation and STEM observation, being considered responsible for its high piezoelectric properties.^{26,37,38}”

Response to Reviewer #3

Comment 1: The current contribution discusses the impact of a multiple of elements on the atomic level structural aspects and the piezoelectric properties of KNN ceramics. It is clearly noticed that the basic underlying idea is an extension from their earlier claims that unit cell level chemical heterogeneity that disturbs a long range polar order and consequently brings about an angular distribution in the direction of polar vector of each unit cell. I see that the collection of experimental data is of high quality, quite complete, and appropriate in supporting their claims, though I think I have to raise a coupling of questions to be clarified.

Reply: We thank the reviewer for his/her positive comments and valuable questions. We will respond to the questions in the following.

Comment 2: First, the presence of the claimed nano-heterogeneity is obvious from their extensive TEM studies, but a demonstration on how these nano-heterogeneity is contributing to the piezoelectricity seems missing. One could assume that the low-angle polar vectors contribute to the piezoelectricity when an electric field is applied through polarization rotation, but this is just a possibility. It could be that or could be a shear contribution from misoriented polar vectors or maybe something else. I do not see a decisive experimental verification on the claimed scenario here.

Reply: We thank the reviewer for the valuable comments. We agree with the reviewer that many potential contributors exist for the enhanced piezoelectricity in multi-elements doped KNN ceramics, including a shear contribution from the misoriented polar vectors.

Based on our microstructure observations, theoretical calculations and property measurements, we are confident that the high piezoelectricity is from the rotation of low-angle polar vectors, because of the following reasons, hope the reviewer is happy with our explanations.

1. The observation of relaxor feature from dielectric measurement indicates the existence of local structure heterogeneity on nanoscale [*Nature*, 546, 391-395 (2017)].
2. There is large amount of local structure heterogeneities with low-angle polar vectors confirmed by STEM.
3. The phase field modeling reveals the contribution from low-angle polar vectors to dielectric/piezoelectric properties is evident, where the interface energy of the local structure heterogeneity with low-angle polar vector is competitive with the Landau energy thus facilitates the polarization rotation. The rotation of polar vectors is actually corresponding to the shear contribution.

Based on the above discussions, we proposed the mechanism, i.e., the dopants induced average tetragonal phase coupled with large amount of local structure heterogeneities with low-angle polar vectors, is responsible for the enhanced

properties in multi-elements doped KNN ceramics.

Comment 3: In fact, their first principles calculation suggested that the coexistence of those multi-elements tends to reduce the anisotropy that is directly proportional to the magnitude of polar vectors. If this is so, the induced angular variation in the direction of polar vectors has a high chance not to make any useful contribution to the piezoelectricity. Please note that the piezoelectricity, especially the electromechanical strain, is a function of both the easy rotation and the magnitude of polar vectors. As noted, one trades off the other.

Reply: Thanks for the valuable comments.

We are sorry about the confusion. The first principles calculation emphasized that the existence of the dopant tended to reduce the bond-length difference between the six B-O bonds in the octahedrons. All the six B-O bonds in the octahedron will decide the magnitude of the polar vectors. We actually observed the overall spontaneous polarization was decreased in KNN-Bi,Sb,Zr comparing to pure KNN. However, the reduced anisotropy of the octahedron will impact the direction of polar vectors, leading to the low-angle polar vectors deviating from the spontaneous polarization direction of the tetragonal phase (confirmed by Synchrotron XRD and STEM). The polar vectors with lower deviation angles are prone to rotate under applied electric field, contributing to the enhanced dielectric permittivity.

We agree with the reviewer that the piezoelectricity and electromechanical strain are a function of both the easy rotation and the magnitude of polar vectors and one trade off the other. According to the equation: $d_{33} \propto \epsilon Q P_s$, where ϵ is dielectric constant (3000 for KNN-Bi,Sb,Zr and 400 for pure KNN) and P_s is spontaneous polarization (20 μC for KNN-Bi,Sb,Zr and 30 μC for pure KNN), while the Q value is electrostrictive coefficient which is insensitive to dopant [*Appl. Phys. Review*, 1, 011103, (2014)], the piezoelectricity and electromechanical strain of KNN-Bi,Sb,Zr are expected to increase because the increase in ϵ is much greater than the decrease in P_s .

Comment 4: In this sense, I think it would be better for the authors to tone down their claim a little by dropping the word “origin” in the title.

Reply: Thanks for the valuable suggestion. We agree with the reviewer that a decisive experimental verification is important on our claimed mechanism, where there are other mechanisms might also contribute to the properties. Following the suggestion, we revised the title as “The mechanism for the enhanced piezoelectricity in multi-elements doped (K_a,Na)NbO₃ ceramics”.

Comment 5: Second, the authors are consistent in emphasizing the effect of multi-elements to get the currently observed enhanced piezoelectricity by comparing the singly doped ones with the triply doped one. I agree that the latter leads to better enhanced piezoelectricity than the former. Then, how about introducing only two different elements? In fact, it has been well known that doubly introduced elements, Li and Sb, are also highly effective in enhancing the piezoelectricity.

Reply: We thank the reviewer for the valuable comments. We agree with the reviewer that the doubly introduced element, Li and Sb, can effectively enhance the piezoelectricity [*J. Appl. Phys.* 100, 104108 (2006)]. The dielectric properties and phase structure of Li and Sb doped KNN were shown below in Fig. R21.

We can see that the KNN-LS5.2 ceramics possess mixed phases at the region of polymorphic phases at room temperature. The maximum d_{33} value of this system is about 260 pC/N.

In this work, we tried to increase the amount of Li and Sb dopants in the KNN system, attempting to introduce the average tetragonal phase in the doped KNN. Unfortunately, secondary phase and impurity were generated due to the solubility limit of these dopants in KNN system, actually each individual dopant in KNN has solubility limitation and cannot be miscible with KNN. On the other hand, if we add the third dopant element, for instance Ta, we can further increase the overall dopant concentration which is enough to induce the average tetragonal phase, in this way, we can increase the piezoelectricity above 400 pC/N.

[Redacted]

Fig. R21 The dielectric properties and XRD patterns of KNN-LS5.2 materials. (a) dielectric permittivity and dielectric loss as a function of temperature and frequency for KNN-LS5.2 materials in -100 – 430°C range. (b) X-ray powder diffraction at different temperatures for sintered KNN-LS5.2 (Figures are from reference of J. Appl. Phys. 100, 104108 (2006), copyright 2006 the American Institute of Physics)

Comment 6: Is the underlying mechanism for the doubly doped KNN different from what is claimed in this work? How about more than three elements?

Reply: Thanks for the good question. The underlying mechanism for the doubly doped KNN is similar to that claimed in this work, but it depends on the solubility of the dopant(s) in the KNN system, as explained in the above.

More than three elements may further increase the overall dopant concentration which is enough to induce the average tetragonal phase, thus resulting in a higher piezoelectricity. There are also publications reporting the doped KNN with more than three dopant elements, as expected, the piezoelectric properties were in the range of 400-550 pC/N [*Acta Mater.* 199, 542-550, (2020)].

Comment 7: Do the authors expect any room for further enhancement in piezoelectricity? In relation to this issue, I am also curious why the authors chose a mixture of A-site substituting Bi and B-site substituting Sb and Zr. Does the proposed mechanism only work with the choice of this type of mixture or also work with any combination of elements?

Reply: Thanks for the valuable comments.

We expect there is still room for further enhancement in piezoelectricity by the

multi-elements doping method. Based on our proposed mechanism, however, we need follow the thumb of rules.

1. A sufficient concentration of dopants is required to induce average tetragonal phase with plenty low-angle polar vectors.
2. The solubility of the dopants has to be considered, ensuring that the impurities and secondary phases are not present.
3. Local structure heterogeneities on both A-site and B-site are thought to benefit the overall microstructure and phase stabilities, considering the tolerance factor and solubility of the dopants.

Based on the above rules, we chose a mixture of A-site Bi^{3+} dopant and B-site $\text{Zr}^{4+}/\text{Sb}^{5+}$ dopants, meanwhile kept the charge neutrality of the whole system.

We expect our proposed mechanism will provide a guideline to design high performance perovskite ceramics employing the minor dopant strategy, not only the combination of Sb/Bi/Zr elements but the combination of chosen elements based on the above thumb of rules.

REVIEWER COMMENTS

Reviewer #1 (Remarks to the Author):

The authors have made substantial changes in the revised manuscript in response to the concerns raised by all three referees. The revised version of the manuscript, in my opinion, is suitable for publication in Nature Communications.

Reviewer #2 (Remarks to the Author):

Gao et al. have revised the manuscript on the mechanism responsible for the enhanced piezoelectricity in multi-element doped KNN-based ceramics. While most of the comments given by the reviewers are addressed in an appropriate manner, there are still a few points which require further revision before the manuscript can be recommended for publication in Nature Comm.

The authors have not addressed the most likely point defects introduced by doping in a satisfactory manner. Still the stoichiometry given in the experimental section is not correct since the nominal cation composition would give oxygen excess (or cation deficiency) or oxygen deficiency for two of the singly doped samples, while for the multi-doped sample the dopants charge compensates for each other and formation of vacancies are not required. This give rise to entirely different dominant point defects in the four different samples, which should be pointed out in the main document. The authors have provided DFT calculations on the B-O bond length, but this does not answer the question regarding the most likely point defect introduced by doping. They also conclude wrongly based on the variation in the B-O bond length (A-site vacancies give more pronounced variation in the bond length not less variation). The DFT data does also show that the B-O bond is indeed sensitive to loss of alkali metal, in contradiction to what the authors conclude.

It is still not clear how the dopants were charge compensated for in the supercell used in the DFT calculations?

The different sintering procedure of the multi-element sample should be pointed out when the grain size is reported. The difference in sintering procedure is important information for the reader.

Reviewer #3 (Remarks to the Author):

The authors responded to all the concerns raised by me during their revision process. As reflected in my earlier review, some of the claims are not satisfactorily answered with the state-of-the-art characterization techniques. The judgment for the feasibility of those issues could now be left for the potential readers in the field.

Wook Jo

Responses to Referees' comments and the description of revisions in the revised manuscript and supplementary materials

We would sincerely thank the referees for their time and efforts in careful reading of the manuscript and in preparation of the review reports. The point-by-point responses to comments were enclosed in the following. We hope we have satisfactorily addressed all referees' concerns and questions. Also the revisions in the manuscript were highlighted.

Response to Reviewer #1

Comment: The authors have made substantial changes in the revised manuscript in response to the concerns raised by all three referees. The revised version of the manuscript, in my opinion, is suitable for publication in Nature Communications.

Reply: We thank the reviewer for his/her comments and the recommendation of publication in Nature Communications. His/Her earlier comments have helped us to improve our manuscript greatly.

Response to Reviewer #2

Comment 1: Gao et al. have revised the manuscript on the mechanism responsible for the enhanced piezoelectricity in multi-element doped KNN-based ceramics. While most of the comments given by the reviewers are addressed in an appropriate manner, there are still a few points which require further revision before the manuscript can be recommended for publication in Nature Comm.

Reply: We are happy that the reviewer thought we had successfully addressed most of the comments, and thank for the constructive comments and suggestions. The point-by-point responses were enclosed in the following and highlighted in the paper.

Comment 2: The authors have not addressed the most likely point defects introduced by doping in a satisfactory manner. Still the stoichiometry given in the experimental section is not correct since the nominal cation composition would give oxygen excess (or cation deficiency) or oxygen deficiency for two of the singly doped samples, while for the multi-doped sample the dopants charge compensates for each other and formation of vacancies are not required. This give rise to entirely different dominant point defects in the four different samples, which should be pointed out in the main document. The authors have provided DFT calculations on the B-O bond length, but this does not answer the question regarding the most likely point defect introduced by doping.

Reply: Thanks for the valuable suggestion, sorry for not clearly listed the point defects in earlier revised version.

For Bi doping element on A-site of KNN, in order to keep the charge balance, the Bi³⁺ donor dopant will induce A-site vacancy, which is the V_k' with negative charge in KNN due to the fact that K₂O possesses low melting temperature of 490°C which may evaporate during high temperature sintering. It should be noted that free electrons with negative charge might also form in Bi doped KNN to keep charge balance, which was not considered in the current research. Meanwhile, for Zr doping element on

B-site of KNN, the Zr^{4+} acceptor dopant will induce oxygen vacancy $V_O^{\cdot\cdot}$ with positive charge to keep the total charge balanced. For the multi-elements doped KNN, on the other hand, the dopants charge compensates for each other and formation of vacancies are not required.

Based on the comment and suggestion, the following paragraph has been added in the main text for describing the point defects introduced by doping in our studied ceramics.

“It should be noted here that Bi doping element on A-site of KNN and Zr doping element on B-site KNN ceramics will introduce point defects, i.e., K vacancy and oxygen vacancy respectively, based on the following defect reactions:

Meanwhile, the Sb doping element on B-site of KNN ceramic will keep charge balanced without introducing the point defects. On the other hand, in our studied multi-elements doped KNN, we attempted to keep the charge balanced by tuning the composition with the appropriate doped levels, thus the formation of point defects is not required.”

We also modified the nominal compositions in Materials Synthesis of the Methods to reflect the charge balance:

We weighed and mixed ... according to the nominal compositions of $K_{0.5}Na_{0.5}Sb_{0.04}Nb_{0.96}O_3$ (KNN-Sb), $K_{0.48}Bi_{0.02}Na_{0.5}NbO_{3.02}$ (KNN-Bi), $K_{0.5}Na_{0.5}Zr_{0.01}Nb_{0.99}O_{2.995}$ (KNN-Zr), and $K_{0.48}Bi_{0.02}Na_{0.5}Nb_{0.92}Sb_{0.04}Zr_{0.04}O_3$ (KNN-Bi,Sb,Zr), respectively.

Comment 3: They also conclude wrongly based on the variation in the B-O bond length (A-site vacancies give more pronounced variation in the bond length not less variation). The DFT data does also show that the B-O bond is indeed sensitive to loss of alkali metal, in contradiction to what the authors conclude.

Reply: We appreciate the valuable comments and are sorry for the confusion.

Based on the DFT data, the dopants and oxygen vacancy will impact the B-O bonds greatly, with bond lengths approaching the same value, while the A-site vacancies slightly change the bond lengths (please kindly check the following Table R1 to compare the B-O bond lengths with pure KNN), with very different lengths of the six B-O bonds, being similar to the pure KNN, demonstrating that the loss of alkali metal has relatively small impact on the octahedrons of KNN.

Table R1. The comparison of the B-O bond lengths of the octahedrons in pure KNN and KNN with K vacancies

	BO-1	BO-2	BO-3	BO-4	BO-5	BO-6
KNN						
	1.883	1.88298	1.97713	1.98182	2.12134	2.12131
25% K-vacancy						
	1.8745	1.90088	1.9886	1.96564	2.12059	2.08279

Based on the above discussions, we revised the sentence in the main text and supporting information for a clear expression.

In the main text:

It should be noted that after adding K vacancy, however, the lengths of the six B-O bonds of the octahedron are still different, though the K vacancy can indeed slightly change the bond lengths of the octahedron comparing to those of the pure KNN (Supplementary Fig. 7), demonstrating that the loss of alkali metal has relatively small impact on the octahedrons of KNN.

In the supporting information (Supplementary Figure 7, Figure caption):

The contributions of the A-site and oxygen vacancies to the structure. The length of the six B-O bonds of KNNs approaches the same value after adding oxygen vacancies, while the difference in the length of the six B-O bonds is still very large by adding A-site vacancy, being similar to the pure KNN.

Comment 4: It is still not clear how the dopants were charge compensated for in the supercell used in the DFT calculations?

Reply: We appreciate the valuable comments and questions.

For single element- doped KNNs, the models were built by substituting K/Na and Nb atoms with in-principle equivalent dopant atoms respectively, i.e., Bi on A-site, Zr on B-site, Sb on B-site. The total charges of the pure and doped KNNs are set to zero, and the charge distribution is automatically balanced in the DFT calculation. This method has been employed in investigations of the properties of doped perovskite oxides, for example, Sm-doped PMN-PT [C. Li, and L. Bellaiche, et al. “*Atomic scale origin of ultrahigh piezoelectricity in samarium doped PMN-PT ceramics,*” *Physical Review B*. 101, 140102 (2020)], Na⁺ doped BiFeO₃ [J. Gebhardt, and A.M. Rappe. “*Accuracy and transferability of Ab initio electronic band structure calculations for doped BiFeO₃,*” *Journal of Physics:Conference Series* 921, 01209 (2017)], Fe and Ni doped PMN-PT [H. Tan, et al., and A.M. Rappe. *First-principles studies of the local structure and relaxor behaviour of Pb(Mg_{1/3}Nb_{2/3})O₃-PbTiO₃-drived ferroelectric perovskite solid solutions.* *Physical Review B*. 97, 174101 (2018)]. Bi doped SnS [Z. Xiao, and F.Y. Ran, et al. “*Route to n-type doping in SnS*” *Applied physical letter* 106, 152103 (2015)], Sr-doped La₂CuO₄ [Furness, J. W. et al. “*An accurate first principles treatment of doping dependent electronic structure of high temperature cuprate superconductors,*” *Communications Physics*. 1, 11 (2018)], Y-doped BaTiO₃ and La, and Sb-doped ASnO₃ (A = Ca, Sr, Ba) [Alshoabi, A., et al. “*Insights into the impact of Yttrium doping at the Ba and Ti sites of BaTiO₃ on the electronic structures and optical properties: A first-principles study,*” *ACS Omega*. 5, 15502-15509 (2020)].

Admittedly it will be reasonable and convincing to consider the charge compensation in the DFT calculation, however, we must use a much larger supercell to consider the vacancies in the Bi- & Zr- doped KNNs, which is very time-consuming. Here, we attempted to perform the DFT calculations for Zr-doped KNN with V_O^{\bullet} located along [001] and [010] directions in the orthorhombic O₆ octahedron. The result is shown in the following Figure R1.

Fig. R1. The lengths of the six B-O bonds of perovskite octahedron for KNN supercell including Zr with oxygen vacancy based on the first-principles calculations.

We can find that the length of the six B-O bonds approaches the same value in Zr-doped KNN considering the oxygen vacancy, this result is similar to that of Zr-doped KNN without considering the charge compensation. Based on the DFT calculations considering the charge compensation, we can conclude a similar evolution trends as in our previous reported results, i.e., the Zr and V_O^\bullet can reduce the difference in the length of the six B-O bonds in oxygen octahedrons. Thus, in the current research, the generally accepted method was used to perform the DFT calculation (The total charge is set to zero, and the charge distribution is automatically balanced in the DFT calculation).

We added the following sentence in the DFT method to give more details about the charge setting.

“It should be noted here that the dopant charge compensation was not considered in the DFT calculations. For the doped KNNs, the models were built by substituting K, Na, and Nb atoms with in-principle equivalent dopant atoms, i.e., Bi on A-site, Sb and Zr on B-site. For the vacancy-defect KNNs, the A-site atom was removed to produce K vacancy, while the corresponding oxygen vacancy was located along [001] and [010] directions in the orthorhombic O_6 octahedron. The total charges of the pure and doped KNNs were set to zero and the charge distribution was automatically balanced in the DFT calculation.”

Comment 5: The different sintering procedure of the multi-element sample should be pointed out when the grain size is reported. The difference in sintering procedure is important information for the reader.

Reply: We appreciate the valuable comments. According to the comment, the sintering procedures for multi-element doped sample and single element doped samples were given in the main text. The revision was also shown in the following:

“The two-step sintering procedure (the details are given in the Materials Synthesis) was employed for the multi-elements doped KNN, where large grain size on the order

of 25 μm was obtained at the optimized sintering temperature, much larger than those of single element doped counterparts prepared by conventional one step sintering process, though all of the samples have a similar density of 94-96%.”

Response to Reviewer #3

Comment 1: The authors responded to all the concerns raised by me during their revision process. As reflected in my earlier review, some of the claims are not satisfactorily answered with the state-of-the-art characterization techniques. The judgment for the feasibility of those issues could now be left for the potential readers in the field.

Reply: We are glad to know that the reviewer agreed to the completeness of the revision process. We appreciate all the constructive comments and suggestions of the reviewer. We believe that the readers will benefit a lot regarding the design of high performance ferroelectric ceramics, from what we have reported in this paper based on the state-of-the-art characterization techniques.

REVIEWERS' COMMENTS

Reviewer #2 (Remarks to the Author):

The authors have made substantial changes in the revised manuscript in response to the concerns raised by all three referees. In the latest revision my final concerns with the manuscript has also been taken into account. The last revised version of the manuscript, in my opinion, is now suitable for publication in Nature Communications.

Responses to Referees' comments

We would sincerely thank the referees for their time and efforts in careful reading of the manuscript and in preparation of the review reports.

Response to Reviewer #2

Comment: The authors have made substantial changes in the revised manuscript in response to the concerns raised by all three referees. In the latest revision my final concerns with the manuscript has also been taken into account. The last revised version of the manuscript, in my opinion, is now suitable for publication in Nature Communications.

Reply: We are happy that the reviewer thought we have address all the concerns of reviewer. We thank the reviewer for his/her comments and the recommendation of publication in Nature Communications.